# Fréchet Geodesic Boosting

**Yidong Zhou**[*]
Department of Statistics
University of California, Davis
Davis, CA 95616
ydzhou@ucdavis.edu

**Su I Iao**[*]
Department of Statistics
University of California, Davis
Davis, CA 95616
siao@ucdavis.edu

**Hans-Georg Müller**
Department of Statistics
University of California, Davis
Davis, CA 95616
hgmueller@ucdavis.edu

## Abstract

Gradient boosting has become a cornerstone of machine learning, enabling base learners such as decision trees to achieve exceptional predictive performance. While existing algorithms primarily handle scalar or Euclidean outputs, increasingly prevalent complex-structured data, such as distributions, networks, and manifold-valued outputs, present challenges for traditional methods. Such non-Euclidean data lack algebraic structures such as addition, subtraction, or scalar multiplication required by standard gradient boosting frameworks. To address these challenges, we introduce *Fréchet geodesic boosting* (FGBoost), a novel approach tailored for outputs residing in geodesic metric spaces. FGBoost leverages geodesics as proxies for residuals and constructs ensembles in a way that respects the intrinsic geometry of the output space. Through theoretical analysis, extensive simulations, and real-world applications, we demonstrate the strong performance and adaptability of FGBoost, showcasing its potential for modeling complex data.

## 1 Introduction

Boosting [52] has emerged as one of the most influential learning paradigms, enabling base learners, such as decision trees, to achieve superior predictive performance. The foundational idea of boosting can be understood as a functional gradient descent method applied to a cost function in function space [6]. Over two decades ago, explicit gradient boosting algorithms were introduced [38, 19], laying the groundwork for their widespread success. Modern iterations of gradient boosting, such as XGBoost [13] and LightGBM [27], have further advanced the field, providing highly efficient and scalable solutions for numerous machine learning tasks.

The increasing availability of complex-structured data in modern science has introduced significant challenges for conventional learning methods [9]. Examples of such data include functional data [59], networks [64], trees [44], distributions [48], and data residing on manifolds such as symmetric positive-definite matrices [46]. These types of data are inherently non-Euclidean and can be viewed as random objects located in metric spaces equipped with suitable metrics. A notable example is the Wasserstein space, where elements are probability distributions and distances are measured using the Wasserstein metric [45]. Despite the success of gradient boosting, existing algorithms are

---

[*]The first two authors contributed equally to this work.

39th Conference on Neural Information Processing Systems (NeurIPS 2025).

predominantly designed for scalar or Euclidean outputs and cannot handle outputs in general metric spaces due to the absence of algebraic operations such as addition or scalar multiplication.

To address this gap, we propose *Fréchet geodesic boosting* (FGBoost), a novel framework designed to adapt gradient boosting for non-Euclidean outputs, specifically for data residing in geodesic metric spaces. FGBoost enables modeling complex regression relationships between Euclidean predictors and non-Euclidean outputs by leveraging the intrinsic geometry of the output space.

## 1.1 Contributions

The primary contributions of this work are as follows:

**Methodology.** We address the challenge of working in geodesic metric spaces, which lack the linear structure required for standard gradient boosting. FGBoost introduces geodesics as proxies for residuals and iteratively constructs an ensemble by adding geodesics while preserving the geometric properties of the output. To achieve this, we develop novel geometric definitions that ensure FGBoost operates intrinsically and adheres to the underlying geometry of the output space. Furthermore, we develop a new version of Shapley Additive Explanations (SHAP) values [37] to enhance the interpretability of FGBoost. To the best of our knowledge, FGBoost represents the first boosting framework designed to effectively accommodate general non-Euclidean outputs.

**Theoretical analysis.** We introduce a general framework to study the theoretical properties of FGBoost, requiring only that the output space is a Hadamard space [56] and demonstrate that the loss function is strongly convex and Lipschitz continuous, guaranteeing the existence and uniqueness of the solution. Using empirical process theory [58], we show that the empirical risk functional converges uniformly to its population counterpart, and the corresponding minimizer is consistent. Detailed proofs of theoretical results are provided in the appendix.

**Simulation studies.** Through extensive numerical experiments, we evaluate the performance of FGBoost across various types of non-Euclidean outputs, including distributions, networks, and compositional data. The results reveal the superiority of FGBoost over existing regression methods designed for non-Euclidean outputs and demonstrate its adaptability to diverse data structures.

**Experiments on real-world data.** We validate the practical utility of FGBoost using real-world datasets from multiple domains. These include distributional data from human mortality studies, networks derived from New York City yellow taxi trip records, and compositional data from a survey of unemployed workers in New Jersey. These applications highlight the ability of FGBoost to effectively model complex data and its relevance across many fields.

## 1.2 Related work

**Gradient boosting.** Recent advancements in gradient boosting have focused on extending its applicability to handle complex-structured data. Examples include algorithms for censored survival data [23, 2, 32], functional data [18, 57, 8], and online learning scenarios [12, 3]. These methods rely on well-defined loss functions (e.g., Cox partial likelihood) or a linear space structure (e.g., Hilbert spaces), enabling adaptation within the gradient boosting framework. Efforts have also been made to incorporate predictive uncertainty, e.g., within the framework of evidential learning [40]. In particular, [40] leverages Wasserstein geometry to construct posterior distributions as intermediate targets for *scalar regression*. By contrast, FGBoost directly learns regression maps into general geodesic spaces, enabling prediction for a wide range of non-Euclidean outputs such as distributions, networks, SPD matrices, and compositional data.

**Regression models for non-Euclidean outputs.** Recent years have seen a surge in regression methods for non-Euclidean outputs. Early approaches include Euclidean embeddings with distance matrices [17] and Nadaraya-Watson kernel regression [22]. More recently, Fréchet regression [47] extended linear and nonparametric regression to metric space-valued outputs. To handle high-dimensional predictors, extensions have incorporated sufficient dimension reduction [60, 62], single index models [4, 20], principal component regression [54], and deep neural networks [25]. However, these methods often depend on restrictive assumptions, such as linear or single-index structures, or low-dimensional manifold constraints. Random forest algorithms have also been adapted for non-Euclidean outputs [10, 49]. In simulations and real-world applications, FGBoost demonstrates superior performance against these alternatives.

## 2 Preliminaries on metric geometry

Let $(\mathcal{M}, d)$ be a bounded metric space. A *curve* in $\mathcal{M}$ is a continuous map $\gamma : [a, b] \to \mathcal{M}$ with length $L(\gamma) = \sup \sum_{i=0}^{I-1} d\{\gamma(t_i), \gamma(t_{i+1})\}$, where the supremum is taken over all possible partitions of the interval $[a, b]$ with arbitrary breakpoints $a = t_0 \leq t_1 \leq \cdots \leq t_I = b$. Two curves $\gamma_1$ and $\gamma_2$ are considered equivalent if there exist non-decreasing, continuous reparametrizations $\phi_1$ and $\phi_2$ such that $\gamma_1 \circ \phi_1 = \gamma_2 \circ \phi_2$. In this case, $\gamma_1$ is said to be a reparametrisation of $\gamma_2$ and one has that $L(\gamma_1) = L(\gamma_2)$. A curve $\gamma : [a, b] \to \mathcal{M}$ is said to have constant speed if for all $a \leq s \leq t \leq b$, $L(\gamma_{[s,t]}) = \frac{t-s}{b-a} L(\gamma)$, where $\gamma_{[s,t]}$ denotes the restriction of $\gamma$ to $[s, t]$. By construction, the metric $d(\alpha, \beta)$ is always less than or equal to the length of any curve connecting $\alpha$ and $\beta$. A metric space $\mathcal{M}$ is called a *length space* if for all $\alpha, \beta \in \mathcal{M}$:

$$d(\alpha, \beta) = \inf_{\gamma} L(\gamma), \tag{1}$$

where the infimum is taken over all curves $\gamma$ connecting $\alpha$ to $\beta$. A length space is a *geodesic space* if for all $\alpha, \beta \in \mathcal{M}$ the infimum on the right-hand side of (1) is attained.

In a geodesic space, a *geodesic* between two points $\alpha$ and $\beta$ is defined as any constant speed curve $\gamma : [0, 1] \to \mathcal{M}$ that achieves the infimum in (1). This geodesic is denoted as $\gamma_{\alpha,\beta}$. If there exists only one such geodesic for all $\alpha, \beta \in \mathcal{M}$, the space $\mathcal{M}$ is a *unique geodesic space* [7].

**Definition 1.** *For $\alpha, \beta, \zeta \in \mathcal{M}$ and $\nu \in [0, 1]$, define the following simple operations on geodesics,*

$$\gamma_{\alpha,\zeta} \oplus \gamma_{\zeta,\beta} := \gamma_{\alpha,\beta}, \ \ominus \gamma_{\alpha,\beta} := \gamma_{\beta,\alpha}, \ \nu \odot \gamma_{\alpha,\beta} = \{\gamma_{\alpha,\beta}(t) : \ t \in [0, \nu]\}, \ \mathrm{id}_{\alpha} := \gamma_{\alpha,\alpha}.$$

These operations generalize the notions of addition, reversal, scalar multiplication, and zero from vectors to geodesics. The following example spaces frequently arise in real-world applications and will feature in our simulations and real-world data applications. Additionally, the space of compositional data is discussed in Appendix A.

**Example 1** (Univariate probability distributions). *Consider the Wasserstein space $(\mathcal{W}, d_{\mathcal{W}})$, which consists of probability distributions on $\mathbb{R}$ with finite second moments, equipped with the Wasserstein metric $d_{\mathcal{W}}$. This space is both complete and separable [45]. The 2-Wasserstein metric between two distributions $\mu_1$ and $\mu_2$ is $d_{\mathcal{W}}^2(\mu_1, \mu_2) = \int_0^1 \{F_{\mu_1}^{-1}(p) - F_{\mu_2}^{-1}(p)\}^2 dp$, where $F_{\mu_1}^{-1}$ and $F_{\mu_2}^{-1}$ are the quantile functions of $\mu_1$ and $\mu_2$, respectively. This space offers a natural framework for analyzing distributions as geometric objects, with geodesics explicitly characterized through optimal transport maps. Denote by $\tau_{\#}\mu$ the pushforward measure of $\mu$ by the transport $\tau$. The geodesic connecting two distributions $\mu_1, \mu_2 \in \mathcal{W}$ is given by McCann's interpolant [41]:*

$$\gamma_{\mu_1,\mu_2}(t) = \{\mathrm{id} + t(F_{\mu_2}^{-1} \circ F_{\mu_1} - \mathrm{id})\}_{\#}\mu_1, \quad t \in [0, 1],$$

*where $\mathrm{id}$ denotes the identity map and $F_{\mu_1}$ is the cumulative distribution function of $\mu_1$.*

**Example 2** (Networks). *Consider the space of simple, undirected, weighted networks with a fixed number of nodes and bounded edge weights. Each network can be represented uniquely by its graph Laplacian. The space of graph Laplacians equipped with the Frobenius metric can thus be used to characterize the space of networks [28, 53, 64]. For any two graph Laplacians $\alpha, \beta$, the geodesic connecting them is the line segment, i.e., $\gamma_{\alpha,\beta}(t) = \alpha + (\beta - \alpha)t$.*

**Example 3** (Symmetric positive-definite matrices). *Consider the space of $l \times l$ symmetric positive-definite matrices $\mathrm{Sym}_l^+$. Common examples include covariance and correlation matrices, which play a crucial role in many statistical and data analysis tasks. Depending on the application, different metrics have been proposed to equip $\mathrm{Sym}_l^+$ with a geometric structure, including the basic Frobenius metric as well as more advanced metrics such as the affine-invariant metric [46], the power metric [15] and the Log-Cholesky metric [33]. Under any of these metrics, $\mathrm{Sym}_l^+$ forms a unique geodesic space.*

## 3 Methodology

### 3.1 Problem formulation

Consider a unique geodesic space $(\mathcal{M}, d)$. Let $(\boldsymbol{X}, Y)$ be a random pair in $\mathbb{R}^p \times \mathcal{M}$. Suppose $\{(\boldsymbol{X}_i, Y_i)\}_{i=1}^n$ form a sample that consists of $n$ independent realizations of $(\boldsymbol{X}, Y)$. FGBoost

presents a novel approach to model the relationship between the non-Euclidean output $Y \in \mathcal{M}$ and a multivariate predictor $\boldsymbol{X} \in \mathbb{R}^p$.

For a random object $Y \in \mathcal{M}$, the Fréchet mean of $Y$, extending the usual notion of mean, is

$$E_{\oplus}(Y) = \underset{\omega \in \mathcal{M}}{\arg\min}\, E\{d^2(Y, \omega)\},$$

where the existence and uniqueness of the minimizer are guaranteed for Hadamard spaces [56] and the example spaces described in Examples 1–3.

## 3.2 Fréchet geodesic boosting

For Euclidean outputs, canonical gradient boosting [19] iteratively constructs an ensemble $F_K$ of $K$ base learners $f_1, \ldots, f_K$, starting with a constant $Y_0$ that best fits the data. At each step $k$, the ensemble is updated as

$$F_1(\boldsymbol{x}) = Y_0 + \nu f_1(\boldsymbol{x}), \quad F_k(\boldsymbol{x}) = F_{k-1}(\boldsymbol{x}) + \nu f_k(\boldsymbol{x}), \quad k = 2, \ldots, K,$$

where $\nu \in (0, 1)$ is a shrinkage parameter, known as the learning rate, which controls the contribution of each base learner to the overall model. Typically, $Y_0$ is chosen as the sample mean of $\{Y_i\}_{i=1}^n$.

The central idea of gradient boosting is to improve the model by iteratively reducing the remaining error of the current ensemble. At each iteration $k$, the base learner $f_k$ is trained to approximate the negative gradient of the loss function at the current prediction $F_{k-1}$. For squared error loss, the negative gradient corresponds to the residuals $Y_i - F_{k-1}(\boldsymbol{X}_i)$. Therefore, the base learner $f_k$ is fitted to the data set $\{(\boldsymbol{X}_i, Y_i - F_{k-1}(\boldsymbol{X}_i))\}_{i=1}^n$. This iterative procedure refines the ensemble, progressively reducing the overall prediction error. Gradient boosting is highly flexible and can employ various base learners, where tree-based models such as decision trees are most common [21].

To explore the regression relationship between non-Euclidean outputs $Y \in \mathcal{M}$ and Euclidean predictors $\boldsymbol{X} \in \mathbb{R}^p$, we propose *Fréchet geodesic boosting* (FGBoost). In the Euclidean setting, the ensemble model is constructed by sequentially adding base learners, each approximating the residual, which corresponds to the negative gradient of the loss function. For unique geodesic spaces, the concept of residuals can be naturally extended to geodesics. Specifically, the residual is replaced by the geodesic connecting the current prediction and the actual observation. Consequently, the ensemble model in FGBoost is defined as the addition of a sequence of geodesics. However, geodesic addition is not inherently well-defined unless the geodesics are connected end to end, preserving continuity. To address this challenge, we introduce the following assumption to ensure well-defined operations within the geodesic framework.

**Assumption 1.** *Let $(\mathcal{M}, d)$ be a unique geodesic space. For any two points $\alpha, \beta \in \mathcal{M}$, there exists a geodesic transport map $T_{\gamma_{\alpha,\beta}} : \mathcal{M} \mapsto \mathcal{M}$ with the following property: $T_{\gamma_{\alpha,\beta}}(\alpha) = \beta$ and for any $\omega \in \mathcal{M}$, there exists a unique point $\zeta \in \mathcal{M}$ such that $T_{\gamma_{\alpha,\beta}}(\omega) = \zeta$.*

This assumption ensures that any geodesic $\gamma_{\alpha,\beta}$ can be naturally extended from any starting point $\omega \in \mathcal{M}$ to a new endpoint $\zeta \in \mathcal{M}$. In the Euclidean space $\mathbb{R}^p$, this map is straightforward and expressed as $T_{\gamma_{\alpha,\beta}}(\omega) = \omega + (\beta - \alpha)$. This construction extends to Hilbert spaces (e.g., $L^2([0, 1])$), where geodesics are straight lines connecting points. An analogous principle can be applied for Riemannian manifolds through parallel transport [61, 35]. Specific definitions of geodesic transport maps for Examples 1–3 are provided in Appendix B. Using the geodesic transport map, we now extend the notion of addition to geodesics that are not connected end to end.

**Definition 2.** *For any points $\alpha, \beta, \omega, \zeta \in \mathcal{M}$ with $\beta \neq \omega$, define the addition between two geodesics $\gamma_{\alpha,\beta}$ and $\gamma_{\omega,\zeta}$ as $\gamma_{\alpha,\beta} \oplus \gamma_{\omega,\zeta} := \gamma_{\alpha,\beta} \oplus \gamma_{\beta,\zeta'} = \gamma_{\alpha,\zeta'}$, where $\zeta' = T_{\gamma_{\omega,\zeta}}(\beta)$.*

The above operation is intuitive in Euclidean space, where $\zeta' = \beta + (\zeta - \omega)$.

For outputs in a unique geodesic space $\mathcal{M}$, the base learner $f_{k+1}$ is trained to approximate the geodesic connecting the current prediction to the actual observation, generalizing the concept of residuals to the geodesic setting. The initial ensemble for FGBoost is defined as $F_0(\boldsymbol{X}_i) = \mathrm{id}_{Y_0}$, which corresponds to the geodesic from a fixed reference point $Y_0 \in \mathcal{M}$ to itself. In practice, $Y_0$ is chosen as the sample Fréchet mean of $\{Y_i\}_{i=1}^n$. The ensemble model is constructed as the addition of a sequence of geodesics, giving rise to the iteration

$$F_k(\boldsymbol{x}) = F_{k-1}(\boldsymbol{x}) \oplus \{\nu \odot f_k(\boldsymbol{x})\}, \quad k = 1, \ldots, K,$$

---
**Algorithm 1** Fréchet Geodesic Boosting
---
**Input:** data $\{(\boldsymbol{X}_i, Y_i)\}_{i=1}^n$, a new predictor level $\boldsymbol{X}$ and a learning rate $\nu \in (0, 1)$.

**Initialize** the model with the estimated Fréchet mean of $\{Y_i\}_{i=1}^n$: $\hat{F}_0(\boldsymbol{x}) = \mathrm{id}_{Y_0}$, where $Y_0 = \arg\min_{\omega \in \mathcal{M}} \frac{1}{n} \sum_{i=1}^n d^2(Y_i, \omega)$.

**for** $k = 1$ **to** $K$ **do**

    **1.** Fit a base learner (e.g. tree) $\hat{f}_k$ to approximate the geodesic from the current prediction to the actual observation using data $\{(\boldsymbol{X}_i, \gamma_{\hat{Y}_i^{k-1}, Y_i})\}_{i=1}^n$, where $\hat{Y}_i^{k-1} = T_{\hat{F}_{k-1}(\boldsymbol{X}_i)}(Y_0)$ denotes the current prediction.

    **2.** Update the ensemble model: $\hat{F}_k(\boldsymbol{x}) = \hat{F}_{k-1}(\boldsymbol{x}) \oplus \{\nu \odot \hat{f}_k(\boldsymbol{x})\}$.

**end for**

**Output:** prediction $\hat{Y} = T_{\hat{F}(\boldsymbol{X})}(Y_0)$ where $\hat{F}(\boldsymbol{X}) := \hat{F}_K(\boldsymbol{X})$.
---

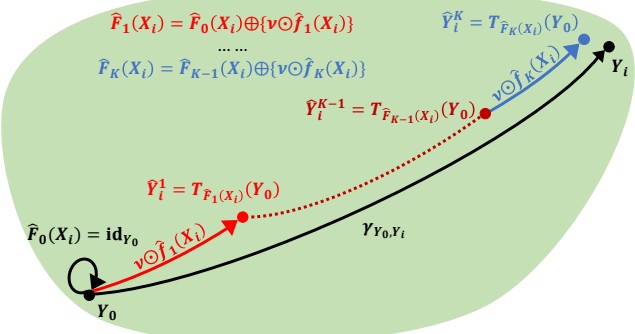

Figure 1: Illustration of the general framework for Fréchet geodesic boosting. The algorithm starts with a fixed reference point $Y_0 \in \mathcal{M}$, serving as the initial estimate. The initial ensemble of Fréchet geodesic boosting is the geodesic from $Y_0$ to itself, $\mathrm{id}_{Y_0}$. At each step, the base learner $\hat{f}_{k+1}$ is trained to approximate the geodesic connecting the current prediction $\hat{Y}_i^k$ and the actual observation $Y_i$. After $K$ iterations, the ensemble model terminates at $\hat{F}_K(\boldsymbol{X}_i)$, and the final prediction for $Y_i$ is $\hat{Y}_K = T_{\hat{F}_K(\boldsymbol{X}_i)}(Y_0)$.

where scalar multiplication and addition of geodesics are defined in Definition 1 and Definition 2, respectively. The ensemble $F_{k-1}(\cdot)$ is the geodesic from $Y_0$ to the current prediction, which can be expressed as $T_{F_{k-1}(\boldsymbol{X}_i)}(Y_0)$ using the geodesic transport map. The complete algorithm for FGBoost is detailed in Algorithm 1, with a schematic illustration provided in Figure 1.

To define a suitable loss function for FGBoost, we need a well-defined metric for comparing geodesics. A geodesic $\gamma_{\alpha, \beta}$ in a unique geodesic space is uniquely characterized by its endpoints $\alpha$ and $\beta$. This observation allows us to represent the space of geodesics as follows:

**Definition 3.** *The space of geodesics on a unique geodesic space* $(\mathcal{M}, d)$ *is*

$$\mathcal{G}(\mathcal{M}) := \{(\alpha, \beta) : \alpha, \beta \in \mathcal{M}\}. \tag{2}$$

Each geodesic $\gamma_{\alpha, \beta}$ is uniquely represented in $\mathcal{G}(\mathcal{M})$ as the pair $(\alpha, \beta)$. To quantify the distance between two geodesics $\gamma_{\alpha_1, \beta_1}, \gamma_{\alpha_2, \beta_2}$, we define the following metric:

$$d_{\mathcal{G}}(\gamma_{\alpha_1, \beta_1}, \gamma_{\alpha_2, \beta_2}) := \sqrt{d^2(\alpha_1, \alpha_2) + d^2(\beta_1, \beta_2)}.$$

**Proposition 1.** $d_{\mathcal{G}}$ *is a valid metric on the space of geodesics* $\mathcal{G}(\mathcal{M})$.

$(\mathcal{G}(\mathcal{M}), d_{\mathcal{G}})$ is thus a metric space, which allows us to formally define the loss function for FGBoost.

FGBoost seeks to find an approximation $\hat{F}$ that minimizes the average loss over the training set,

$$\hat{F} = \arg\min_F \frac{1}{n} \sum_{i=1}^n d_{\mathcal{G}}^2(\gamma_{Y_0, Y_i}, F(\boldsymbol{X}_i)),$$

where $F$ maps the predictor $\boldsymbol{X}_i$ to a geodesic connecting $Y_0$ and the prediction. FGBoost starts with an initial constant model $\hat{F}_0$ and incrementally adds base learners in a greedy manner. At iteration $k$, the base learner is trained as

$$\hat{f}_k = \arg\min_{f_k} \frac{1}{n} \sum_{i=1}^{n} d^2(Y_i, T_{\hat{F}_{k-1}(\boldsymbol{X}_i) \oplus \{\nu \odot f_k(\boldsymbol{X}_i)\}}(Y_0)), \tag{3}$$

where $\hat{F}_{k-1}(\boldsymbol{X}_i)$ is the current prediction and $T_{\hat{F}_{k-1}(\boldsymbol{X}_i) \oplus \{\nu \odot f(\boldsymbol{X}_i)\}}(Y_0)$ represents the ending point of the updated prediction after incorporating the new base learner.

Although the expression for $\hat{f}_k$ in (3) may appear to involve a nested optimization, in practice FGBoost adopts the same greedy approximation used in classical gradient boosting. This reduces the task to fitting decision trees to pseudo-residuals. At iteration $k$, we compute the geodesics $\gamma_{\hat{Y}_i^{k-1}, Y_i}$ connecting the current predictions $\hat{Y}_i^{k-1}$ to the observed responses $Y_i$ and treat these geodesics as pseudo-residuals. A decision tree is then trained on the pairs $(\boldsymbol{X}_i, \gamma_{\hat{Y}_i^{k-1}, Y_i})$, providing a tractable approximation to the idealized optimization problem.

Tree construction follows the standard greedy procedure of decision trees. For any candidate split defined by a feature and threshold, the data are divided into two child regions $R_1$ and $R_2$. Each region is assigned a representative geodesic obtained as the Fréchet mean of the pseudo-residuals it contains:

$$\gamma_j = \arg\min_{\gamma_{\omega', \omega} \in \mathcal{G}(\mathcal{M})} \sum_{i: \boldsymbol{X}_i \in R_j} d_{\mathcal{G}}^2 \left( \gamma_{\hat{Y}_i^{k-1}, Y_i}, \gamma_{\omega', \omega} \right), \quad j \in \{1, 2\}.$$

Under the metric $d_{\mathcal{G}}$, this optimization decouples into two simpler Fréchet mean problems: one over the starting points $\{\hat{Y}_i^{k-1}\}$ and one over the endpoints $\{Y_i\}$. The resulting $\gamma_j$ is therefore the geodesic connecting the two means. This decoupling is purely a computational device for efficient leaf estimation and does not compromise the geometric integrity of the method.

The quality of a candidate split is measured by the resulting mean squared error, i.e., the sum of squared distances between the observed responses $Y_i$ and their updated predictions after applying the representative geodesics. The split that yields the greatest reduction in this loss is selected, and the process is repeated recursively until a stopping criterion (e.g., maximum depth) is met.

## 4 Theoretical analysis

We introduce a general framework to study the theoretical properties of FGBoost. Let $\mathcal{F}$ represent the class of base learners $f : \mathbb{R}^p \to \mathcal{G}(\mathcal{M})$, where $\mathcal{G}(\mathcal{M})$ is the space of geodesics as per (2). Define span$(\mathcal{F})$ as the set of all linear combinations of base learners in $\mathcal{F}$. For any $F \in \text{span}(\mathcal{F})$, one has $F(\boldsymbol{x}) = \text{id}_{Y_0} \oplus \{\nu_1 \odot f_1(\boldsymbol{x})\} \oplus \cdots \oplus \{\nu_K \odot f_K(\boldsymbol{x})\}$, where $Y_0$ is a fixed reference point and $f_k \in \mathcal{F}$ for $k = 1, \ldots, K$. Let $\psi : \mathcal{G}(\mathcal{M}) \times \mathcal{G}(\mathcal{M}) \to [0, \infty)$ denote the loss function, defined as $\psi(\gamma, F) = d_{\mathcal{G}}^2(\gamma, F)$, where $d_{\mathcal{G}}$ is the metric on the space of geodesics $\mathcal{G}(\mathcal{M})$. FGBoost aims to construct a function $F : \mathbb{R}^p \to \mathcal{G}(\mathcal{M})$ that minimizes the empirical risk functional $A_n(F) = \frac{1}{n} \sum_{i=1}^{n} \psi(\gamma_{Y_0, Y_i}, F(\boldsymbol{X}_i))$. The population counterpart of this risk functional is $A(F) = E\{\psi(\gamma_{Y_0, Y}, F(\boldsymbol{X}))\}$.

**Remark 1.** *Throughout this section, we assume that $(M, d)$ is a bounded metric space, as introduced in Section 2. For spaces that are potentially unbounded, in practice the data distribution is typically supported on a stochastically bounded subset, so the diameter can be taken as a high-probability bound. This boundedness assumption is standard in the analysis of metric space-valued data and is reasonable for practical applications [16, 30].*

To ensure that the optimization problem is well-posed, it is crucial that the loss function $\psi$ satisfies certain desirable properties. These properties are guaranteed when $\mathcal{M}$ is a Hadamard space.

**Definition 4** (Hadamard space). *A metric space $(\mathcal{M}, d)$ is a Hadamard space if it is complete and if for each pair of points $\omega_1, \omega_2 \in \mathcal{M}$ there exists a point $\alpha \in \mathcal{M}$ with the property that for all points $\beta \in \mathcal{M}$:*

$$d^2(\beta, \alpha) \leq \frac{1}{2} d^2(\beta, \omega_1) + \frac{1}{2} d^2(\beta, \omega_2) - \frac{1}{4} d^2(\omega_1, \omega_2).$$

Hadamard spaces, also known as global NPC (Non-Positive Curvature) spaces [56], are unique geodesic spaces. The spaces discussed in Examples 1–3 fall within this category. The following proposition establishes key properties of the loss function $\psi$ in a Hadamard space.

**Proposition 2.** *If $(\mathcal{M}, d)$ is a Hadamard space, then for any geodesic $\gamma \in \mathcal{G}(\mathcal{M})$, the function $\psi(\gamma, \cdot)$ is strongly convex over $\mathcal{G}(\mathcal{M})$ and Lipschitz continuous with respect to $d_{\mathcal{G}}$.*

The strong convexity and continuity of $\psi$ enable us to establish the existence and uniqueness of a solution for the risk minimization problem. These properties are crucial for analyzing the asymptotic behavior of gradient boosting algorithms; see for example [63].

**Theorem 1.** *If $(\mathcal{M}, d)$ is a Hadamard space, then the optimization problems $\arg\min_{F \in span(\mathcal{F})} A(F)$ and $\arg\min_{F \in span(\mathcal{F})} A_n(F)$ each admit a unique solution.*

To address the challenge posed by the absence of linear operations, we employ tools from empirical process theory [58] to study the asymptotic behavior of the empirical risk functional $A_n(F)$ and establish that $A_n(F)$ converges uniformly to the population risk functional $A(F)$ over $F \in \mathcal{F}$ as the sample size grows, which then guarantees the convergence of the corresponding minimizer.

**Theorem 2.** *Suppose $(\mathcal{M}, d)$ is a Hadamard space. Then $\sup_{F \in span(\mathcal{F})} |A_n(F) - A(F)| = o_p(1)$. Furthermore, $\sup_{\boldsymbol{x} \in \mathbb{R}^p} d_{\mathcal{G}}(F_n^*(\boldsymbol{x}), F^*(\boldsymbol{x})) = o_p(1)$, where $F_n^* = \arg\min_{F \in span(\mathcal{F})} A_n(F)$ and $F^* = \arg\min_{F \in span(\mathcal{F})} A(F)$.*

## 5   Numerical experiments

We assess the performance of FGBoost through comprehensive numerical simulations involving non-Euclidean outputs, specifically distributional data modeled in the Wasserstein space equipped with the Wasserstein metric, and network data represented by graph Laplacians using the Frobenius metric, as detailed in Examples 1 and 2. Simulations are conducted with sample sizes of $n = 100, 200, 500, 1000, 2000$, and each scenario is replicated across 500 runs. FGBoost is benchmarked against state-of-the-art regression models for non-Euclidean outputs, including global Fréchet regression (GFR) [47], sufficient dimension reduction (SDR) [62], single index Fréchet regression (IFR) [4], Fréchet random forest (FRF) [10], and random forest weighted local linear Fréchet regression (RFWLLFR) [49]. Due to their high computational cost, SDR and IFR are not evaluated at sample size $n = 2000$. A detailed comparison of training times across all models is provided in Appendix J. Additional simulations for compositional data are presented in Appendix A. Code for implementing FGBoost is available at `https://github.com/SUIIAO/FGBoost`.

**Common hyperparameters.**   In all simulations, the learning rate $\nu$ is set to 0.05, and the number of iterations $K$ is fixed at 100. The depth of the tree is fixed at 3, with each leaf requiring a minimum of 10 samples. Tuning these parameters can be accomplished through a grid search, assessing empirical risk with cross-validation. Additionally, 10% of the training set is reserved as the validation set in each run. The training process halts when the empirical risk on the validation set no longer shows consistent improvement.

**Performance evaluation.**   The out-of-sample performance of FGBoost is assessed using the mean squared prediction error (MSPE). Write $m(\cdot)$ for the true regression function and $\hat{m}^q(\cdot)$ for the fitted regression function for the $q$th Monte Carlo run. The MSPE is computed as $\text{MSPE}_q = \frac{1}{100} \sum_{i=1}^{100} d^2\{\hat{m}_q(\boldsymbol{X}_i^{\text{test}}), m(\boldsymbol{X}_i^{\text{test}})\}$, where $\{\boldsymbol{X}_i^{\text{test}}\}_{i=1}^{100}$ denote out-of-sample predictors and $d$ is the metric for the corresponding metric space. The average performance over 500 Monte Carlo runs is quantified by $\text{AMSPE} = \frac{1}{500} \sum_{q=1}^{500} \text{MSPE}_q$.

**Distributions.**   We consider truncated one-dimensional Gaussian distributions with random parameters $\eta$ and $\sigma$ on $[-2, 2]$ as distributional outputs, characterized by quantile functions $Q(p) = E(\eta|\boldsymbol{X}) + E(\sigma|\boldsymbol{X})\Phi^{-1}(\Phi(a) + p\{\Phi(b) - \Phi(a)\})$, where $\Phi(\cdot)$ represents the cumulative distribution function of the standard Gaussian distribution, $a = \frac{-2 - E(\eta|\boldsymbol{X})}{E(\sigma|\boldsymbol{X})}$ and $b = \frac{2 - E(\eta|\boldsymbol{X})}{E(\sigma|\boldsymbol{X})}$. To generate distributional outputs, the predictor $\boldsymbol{X} \in \mathbb{R}^9$ is sampled as follows:

$$X_1 \sim U(0,1),\ X_2 \sim U(-1,1),\ X_3 \sim U(-2,2),\ X_4 \sim N(0,1),\ X_5 \sim N(0,1),$$
$$X_6 \sim N(0,1),\ X_7 \sim \text{Ber}(0.1),\ X_8 \sim \text{Ber}(0.2),\ X_9 \sim \text{Ber}(0.5).$$

Mean $\eta$ and standard deviation $\sigma$ of the truncated Gaussian distribution that serves as the distributional output are generated conditional on predictor vectors $\boldsymbol{X}$, where

$$\eta|\boldsymbol{X} \sim N(\mu, 0.5^2),\ \sigma|\boldsymbol{X} \sim \text{Gamma}(\theta^2, \theta^{-1})\ \text{and}$$
$$\mu = \sin(\pi X_1) - \cos(\pi X_4)X_7,\ \theta = 1 + 2\cos(\pi X_2/2) + X_5^2 X_8.$$

To mimic real-world scenarios where direct access to probability distributions is unavailable, we simulate independent data samples for each distributional output. Specifically, 100 observations $\{y_{ij}\}_{j=1}^{100}$ are sampled independently from each distribution $Y_i$. Consequently, one must initially estimate the distributional output $Y_i$ from the random sample $\{y_{ij}\}_{j=1}^{100}$, introducing a bias in the regression model. This setup reflects practical challenges and aligns with prior approaches [65], where the empirical measure is adopted as a proxy for the latent distribution $Y_i$.

**Networks.** Consider simple, undirected, weighted networks with a fixed number of nodes $l$ and bounded edge weights. Such networks can be uniquely characterized using their graph Laplacians [64], which are symmetric matrices satisfying specific constraints. Formally, the space of graph Laplacians is

$$\mathcal{M} = \{Y = (y_{ij}) : Y = Y^{\text{T}}; Y\mathbf{1}_l = \mathbf{0}_l;\ \text{there exists } W > 0 \text{ such that } -W \leq y_{ij} \leq 0 \text{ for } i \neq j\},$$

where $\mathbf{1}_l$ and $\mathbf{0}_l$ are $l$-vectors of ones and zeroes, respectively.

To construct a generative model for network outputs, we draw predictors $\boldsymbol{X} \in \mathbb{R}^9$ from the following distributions:

$$X_1 \sim U(-1,1),\ X_2 \sim U(-1,1),\ X_3 \sim U(1,2),\ X_4 \sim \text{Gamma}(3,1),\ X_5 \sim \text{Gamma}(4,1),$$
$$X_6 \sim \text{Gamma}(5,1),\ X_7 \sim \text{Ber}(0.2),\ X_8 \sim \text{Ber}(0.3),\ X_9 \sim \text{Ber}(0.5).$$

For the corresponding network output, the weights of the edges are modeled using a beta distribution with shape parameters $\alpha = 2X_7 \sin^2(\pi X_1) + (1 - X_7)\cos^2(\pi X_2)$ and $\beta = X_4^2 X_8 + X_5^2(1 - X_8)$. The generated edge weights are then used to construct the graph Laplacian, which serves as the output. As an alternative to the proposed geometry-aware approach, one could apply XGBoost [13] to vectorized representations of the graph Laplacians. We evaluate this baseline approach in Appendix H and find that it underperforms FGBoost, highlighting the importance of respecting the geometric structure of the output space.

**Discussion on the simulation results.** Table 1 presents the AMSPE for FGBoost and the competing models. As the sample size increases, FGBoost exhibits a clear trend of decreasing prediction error, indicating its convergence to the target. Across all data types and sample sizes considered, FGBoost consistently outperforms the competing methods. This performance gap becomes increasingly pronounced with larger sample sizes, underscoring the scalability and accuracy of FGBoost in capturing complex relationships between multivariate predictors and non-Euclidean outputs.

# 6  Real-world data applications

We evaluate the performance of the FGBoost algorithm using real-world datasets, including human mortality data with distributional outputs and New York City yellow taxi data with network outputs. An additional application involving compositional data is presented in the appendix. To assess predictor importance, we developed an adapted version of Shapley Additive Explanations (SHAP) values [37], with further details provided in the appendix.

## 6.1  Human mortality data

This analysis uses age-at-death distributions from 162 countries in 2015 as distributional outputs. The life tables, sourced from the United Nations World Population Prospects 2024 (`https://population.un.org/wpp/downloads`), provide death counts grouped into five-year age intervals. Using the `frechet` package [14], these aggregated death counts were smoothed via a local linear smoother and then normalized through trapezoidal integration to generate density estimates. Age-at-death distributions are influenced by a variety of factors, and we consider a nine-dimensional predictor

Table 1: Average mean squared prediction errors and standard deviations (in parentheses) of FGBoost, global Fréchet regression (GFR) [47], sufficient dimension reduction (SDR) [62], single index Fréchet regression (IFR) [4], Fréchet random forest (FRF) [10] and random forest weighted local linear Fréchet regression (RFWLLFR) [49] for simulated distributional and network outputs.

| Output | $n$ | FGBoost | GFR | SDR | IFR | FRF | RFWLLFR |
|---|---|---|---|---|---|---|---|
| Distribution | 100 | **0.034** | 0.053 | 0.098 | 0.041 | 0.048 | 0.097 |
| | | (0.011) | (0.014) | (0.043) | (0.012) | (0.013) | (0.042) |
| | 200 | **0.028** | 0.043 | 0.059 | 0.038 | 0.041 | 0.075 |
| | | (0.010) | (0.009) | (0.022) | (0.008) | (0.010) | (0.021) |
| | 500 | **0.023** | 0.038 | 0.040 | 0.037 | 0.034 | 0.057 |
| | | (0.009) | (0.008) | (0.013) | (0.008) | (0.008) | (0.013) |
| | 1000 | **0.019** | 0.037 | 0.035 | 0.037 | 0.029 | 0.046 |
| | | (0.007) | (0.007) | (0.012) | (0.008) | (0.006) | (0.009) |
| | 2000 | **0.015** | 0.036 | — | — | 0.026 | 0.038 |
| | | (0.005) | (0.006) | — | — | (0.005) | (0.006) |
| Network | 100 | **13.644** | 15.326 | 19.448 | 17.768 | 13.820 | 19.321 |
| | | (3.140) | (2.570) | (11.786) | (5.529) | (2.743) | (4.157) |
| | 200 | **10.531** | 14.309 | 14.391 | 16.619 | 12.190 | 16.528 |
| | | (3.371) | (2.474) | (2.862) | (3.325) | (2.372) | (3.113) |
| | 500 | **6.912** | 13.831 | 12.473 | 16.382 | 10.659 | 14.572 |
| | | (1.950) | (2.591) | (2.351) | (3.343) | (2.205) | (2.769) |
| | 1000 | **5.471** | 13.481 | 11.769 | 16.100 | 9.376 | 13.066 |
| | | (1.481) | (2.383) | (1.865) | (2.979) | (1.957) | (2.673) |
| | 2000 | **4.996** | 13.459 | — | — | 8.308 | 11.891 |
| | | (1.325) | (2.329) | — | — | (1.765) | (2.450) |

Table 2: Average mean squared prediction errors and standard deviations (in parentheses) of FGBoost, global Fréchet regression (GFR) [47], sufficient dimension reduction (SDR) [62], single index Fréchet regression (IFR) [4], Fréchet random forest (FRF) [10] and random forest weighted local linear Fréchet regression (RFWLLFR) [49] for human mortality and taxi network data.

| Data | FGBoost | GFR | SDR | IFR | FRF | RFWLLFR |
|---|---|---|---|---|---|---|
| Human Mortality | **20.35** | 31.41 | 27.60 | 58.36 | 22.02 | 22.55 |
| | (36.63) | (58.83) | (44.40) | (107.54) | (38.08) | (35.49) |
| Taxi Network | **8.30** | 11.80 | 13.15 | 32.16 | 9.50 | 10.17 |
| | (0.18) | (0.20) | (0.38) | (2.14) | (0.15) | (0.68) |

set encompassing demographic, economic, and environmental variables. Detailed descriptions of these predictors are provided in Table 9 in the appendix.

To evaluate model performance, leave-one-out cross-validation is employed to compute the MSPE, with results summarized in Table 2. FGBoost achieves the lowest MSPE, demonstrating its effectiveness in capturing complex distributional relationships, even with relatively small sample sizes. Figure 2 presents the SHAP summary plot, ranking predictors by their importance in FGBoost. The analysis reveals that GDP has the most substantial impact on age-at-death distributions, corroborating prior studies emphasizing the critical role of socioeconomic factors in health outcomes [43, 42]. Other influential factors include mean childbearing age, health expenditure, and population density.

## 6.2 New York City yellow taxi data

We analyze transport dynamics in Manhattan, New York, using yellow taxi trip records obtained from https://www.nyc.gov/site/tlc/about/tlc-trip-record-data.page. Manhattan is partitioned into 13 regions, for which daily transport networks are constructed. The nodes of the networks represent the 13 regions and their edge weights indicate the number of passengers traveling between them. We characterize these networks by $13 \times 13$ graph Laplacian matrices and associate these with 12-dimensional predictor vectors, with components including weather attributes and weekday/holiday indicators, as detailed in Table 10 in the appendix. We examine how these factors impact the transport networks, based on data spanning January 1, 2017, to December 31, 2018.

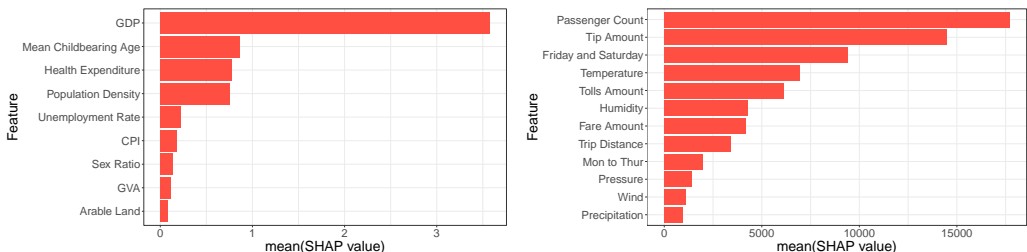

Figure 2: Summary plot of SHAP values for FGBoost applied to human mortality data (left) and taxi network data (right). Features are sorted by their impact in descending order.

Model performance is evaluated through five-fold cross-validation, with the MSPE averaged over 100 runs, as reported in Table 2. FGBoost achieves the lowest MSPE, consistently outperforming all competing models in predicting transport networks. Figure 2 displays the SHAP summary plot, ranking predictors by their influence on the predictions obtained by FGBoost. Key factors include passenger count, tip amount, and the indicator for whether it is a Friday/Saturday, with temperature and toll amounts also contributing, though to a lesser extent.

# 7    Conclusion

We propose FGBoost, an innovative intrinsic regression method designed for geodesic metric spaces that successfully addresses the challenge of the absence of linear operations, which are essential for traditional gradient boosting. By leveraging geodesics as proxies for residuals, FGBoost iteratively builds ensembles while preserving the geometric properties of the output space, ensuring intrinsic compatibility with the underlying metric space structure. FGBoost is supported by theoretical results and its performance is highly competitive in numerical experiments and real data analysis.

Future research could focus on extending the theory of FGBoost beyond Hadamard spaces. Another extension of interest for future research will be to extend FGBoost to handle scenarios where both predictors and outputs reside in general metric spaces. One potential solution involves modifying the splitting rules for tree-based FGBoost to operate intrinsically within metric spaces [10]. This would enable FGBoost to address regression tasks with non-Euclidean predictors.

# Acknowledgments and Disclosure of Funding

We would like to thank the reviewers for their constructive feedback. This research was partially supported by NSF grant DMS-2310450.

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

# A Compositional data

In this section, we explore another important example: the space of compositional data. We demonstrate the applicability and effectiveness of FGBoost for this type of data through numerical simulations and a real-world case study.

**Example 4** (Space of compositional data). *Compositional data, represented as proportions summing to 1, reside in the simplex: $\Delta^{d-1} = \{\boldsymbol{y} \in \mathbb{R}^d : y_j \geq 0, j = 1, \ldots, d, \text{ and } \sum_{j=1}^d y_j = 1\}$. Using the square-root transformation $\sqrt{\boldsymbol{y}} = (\sqrt{y_1}, \ldots, \sqrt{y_d})^{\mathrm{T}}$, the simplex can be mapped to the positive orthant of the unit sphere [50, 51]: $\mathcal{S}_+^{d-1} = \{\boldsymbol{z} \in \mathcal{S}^{d-1} : z_j \geq 0, j = 1, \ldots, d\}$. Equipping $\mathcal{S}_+^{d-1}$ with the geodesic (Riemannian) metric on the sphere, $d_g(\boldsymbol{z}_1, \boldsymbol{z}_2) = \arccos(\boldsymbol{z}_1^{\mathrm{T}} \boldsymbol{z}_2)$ for $\boldsymbol{z}_1, \boldsymbol{z}_2 \in \mathcal{S}_+^{d-1}$, induces a unique geodesic structure. The geodesic connecting two points $\boldsymbol{z}_1, \boldsymbol{z}_2 \in \mathcal{S}_+^{d-1}$ is explicitly defined as:*

$$\gamma_{\boldsymbol{z}_1, \boldsymbol{z}_2}(t) = \cos(t\theta)\boldsymbol{z}_1 + \sin(t\theta)\frac{\boldsymbol{z}_2 - (\boldsymbol{z}_1^{\mathrm{T}} \boldsymbol{z}_2)\boldsymbol{z}_1}{\|\boldsymbol{z}_2 - (\boldsymbol{z}_1^{\mathrm{T}} \boldsymbol{z}_2)\boldsymbol{z}_1\|}, \quad t \in [0, 1],$$

*where $\theta = \arccos(\boldsymbol{z}_1^{\mathrm{T}} \boldsymbol{z}_2)$ is the angle between $\boldsymbol{z}_1$ and $\boldsymbol{z}_2$.*

## A.1 Experiments for compositional data

Consider three-dimensional compositional data residing on $\mathcal{S}_+^2$, the positive segment of the unit sphere in $\mathbb{R}^3$, equipped with the geodesic metric. The Euclidean predictor $\boldsymbol{X} \in \mathbb{R}^{10}$ includes $X_1, \ldots, X_9$, distributed identically to the predictors in the network simulation, and an additional variable $X_{10} \sim \text{Ber}(0.1)$.

The true regression function is modeled as as $m(\boldsymbol{X}) = (1 - X_8)m_0(\boldsymbol{X}) + X_8 m_1(\boldsymbol{X})$, where

$$m_0(\boldsymbol{X}) = (\cos(\phi), \sqrt{3}\sin(\phi)/2, \sin(\phi)/2), \quad m_1(\boldsymbol{X}) = (\cos(\phi), \sin(\phi)/2, \sqrt{3}\sin(\phi)/2),$$

and $\phi = \pi(f(\boldsymbol{X}) + 2)/8 \in [\pi/8, 3\pi/8]$. The function $f(\boldsymbol{X})$ is defined as $f(\boldsymbol{X}) = b(\boldsymbol{X})/\{|a(\boldsymbol{X})| + |b(\boldsymbol{X})|\}$ where $a(\boldsymbol{X}) = 3X_{10}\sin^2(\pi X_1) + 3(1 - X_{10})\cos^2(\pi X_2)$ and $b(\boldsymbol{X}) = -X_7\sqrt{X_4} + (1 - X_7)\sqrt{X_5}$.

To generate the random output $Y$ on $\mathcal{S}_+^2$ for $X_8 = 0$, we add a small perturbation to the true regression function $m_0(\boldsymbol{X})$. First, we construct an orthonormal basis $(e_1, e_2)$ for the tangent space at $m_0(\boldsymbol{X})$ where

$$e_1 = (\sin(\phi), -\sqrt{3}\cos(\phi)/2, -\cos(\phi)/2), \quad e_2 = (0, 1/2, -\sqrt{3}/2).$$

Next, we consider random tangent vectors $U = Z_1 e_1 + Z_2 e_2$, where $Z_1, Z_2$ are independent random variables uniformly distributed on $[-0.1, 0.1]$. The random output $Y$ is then obtained by applying the exponential map at $m_0(\boldsymbol{X})$ to $U$,

$$Y = \text{Exp}_{m_0(\boldsymbol{X})}(U) = \cos(\|U\|)m_0(\boldsymbol{X}) + \sin(\|U\|)\frac{U}{\|U\|}.$$

For $X_8 = 1$, a similar procedure is followed with the orthonormal basis $(e_1, e_2)$ for the tangent space at $m_1(\boldsymbol{X})$ defined as

$$e_1 = (\sin(\phi), -\cos(\phi)/2, -\sqrt{3}\cos(\phi)/2), \quad e_2 = (0, \sqrt{3}/2, -1/2).$$

Figure 3 illustrates randomly generated outputs using the above generation procedure for a sample size $n = 500$.

FGBoost consistently outperforms competing methods across a range of sample sizes, as shown in Table 3. The sole exception occurs at a smaller sample size of $n = 100$, where GFR and FRF exhibit slightly better performance. However, as the sample size grows, FGBoost not only catches up but also showcases a marked improvement, further solidifying its superiority in handling larger datasets.

## A.2 Emotional well-being for unemployed workers

A survey of unemployed workers in New Jersey [29] was conducted during the fall of 2009 and early 2010, a period when the U.S. unemployment rate peaked at 10% following the 2007–2008 financial crisis. The analysis includes data for $n = 3301$ unemployed workers with complete measurements.

Table 3: Average mean squared prediction errors and standard deviations (in parentheses) of Fréchet geodesic boosting (FGBoost), global Fréchet regression (GFR) [47], sufficient dimension reduction (SDR) [62], single index Fréchet regression (IFR) [4], Fréchet random forest (FRF) [10] and random forest weighted local linear Fréchet regression (RFWLLFR) [49] for compositional outputs.

| Output | $n$ | FGBoost | GFR | SDR | IFR | FRF | RFWLLFR |
|---|---|---|---|---|---|---|---|
| | 100 | 0.0104 | 0.0099 | 0.0382 | 0.0800 | **0.0088** | 0.0456 |
| | | (0.0041) | (0.0012) | (0.0068) | (0.0082) | (0.0012) | (0.0639) |
| | 200 | **0.0074** | 0.0089 | 0.0368 | 0.0782 | 0.0076 | 0.0243 |
| | | (0.0018) | (0.0009) | (0.0058) | (0.0078) | (0.0011) | (0.0302) |
| Compositional | 500 | **0.0047** | 0.0085 | 0.0364 | 0.0780 | 0.0064 | 0.0112 |
| | | (0.0008) | (0.0008) | (0.0057) | (0.0073) | (0.0009) | (0.0017) |
| | 1000 | **0.0038** | 0.0083 | 0.0361 | 0.0770 | 0.0056 | 0.0092 |
| | | (0.0007) | (0.0008) | (0.0054) | (0.0077) | (0.0009) | (0.0014) |

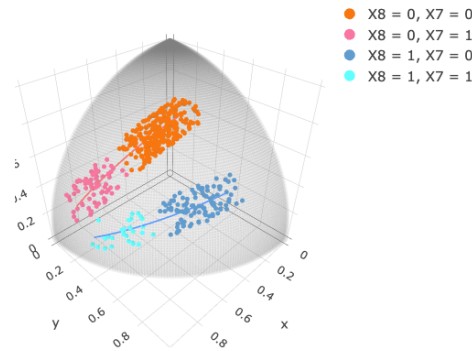

Figure 3: Visualization of simulated compositional data for $n = 500$.

The key response variable is the proportion of time spent in each of the four moods while at home: bad, low/irritable, mildly pleasant, and very good. The compositional response vector is represented as $\boldsymbol{y} = (y_1, y_2, y_3, y_4)^{\mathrm{T}}$, where $y_j$ denotes the proportion of time spent in the $j$th mood ($j = 1, \ldots, 4$). Applying a square-root transformation, $\boldsymbol{z} = (z_1, z_2, z_3, z_4)^{\mathrm{T}} = \left(\sqrt{y_1}, \sqrt{y_2}, \sqrt{y_3}, \sqrt{y_4}\right)^{\mathrm{T}}$, maps the outputs to the positive orthant of the unit sphere $\mathcal{S}_+^3$. The corresponding geodesic metric $d(\boldsymbol{z}_1, \boldsymbol{z}_2) = \arccos(\boldsymbol{z}_1^{\mathrm{T}} \boldsymbol{z}_2)$ is used for the analysis. The predictors for this application consist of 10 baseline socio-economic and demographic variables collected through the questionnaire: (1) life satisfaction (2) highest education level (3) marital status (4) the number of children, (5) the number of people in the household, (6) total annual household income, (7) hours per week working at the last job, (8) how the last job ended, (9) weeks spent looking for work, and (10) credit card balance.

Model performance is assessed using ten-fold cross-validation, with the MSPE averaged over 100 runs, as reported in Table 4. FGBoost demonstrates a substantial improvement, achieving more than a 50% reduction in MSPE compared to GFR, SDR, and IFR. FRF and RFWLLFR are not included in the comparison as their implementations only work for three-dimensional compositional data. Figure 4 presents the SHAP summary plot, ranking predictors by their importance. The analysis identifies life satisfaction as the most influential predictor, followed by credit balance, weeks spent job seeking, and household size, though their impacts are considerably smaller than that of life satisfaction.

## B  Geodesic transport maps

In the Wasserstein space from Example 1, Assumption 1 is satisfied with the geodesic transport map $T_{\gamma_{\alpha,\beta}} = F_\beta^{-1} \circ F_\alpha$, where $F_\alpha$ and $F_\beta^{-1}$ are the cumulative distribution function of $\alpha$ and

Table 4: Average mean squared prediction errors and standard deviations (in parentheses) of Fréchet geodesic boosting (FGBoost), global Fréchet regression (GFR) [47], sufficient dimension reduction (SDR) [62], and single index Fréchet regression (IFR) [4] for emotional well-being data.

| FGBoost | GFR | SDR | IFR |
|---|---|---|---|
| **0.2074 (0.0005)** | 0.4163 (0.0007) | 0.4112 (0.0007) | 0.4356 (0.0015) |

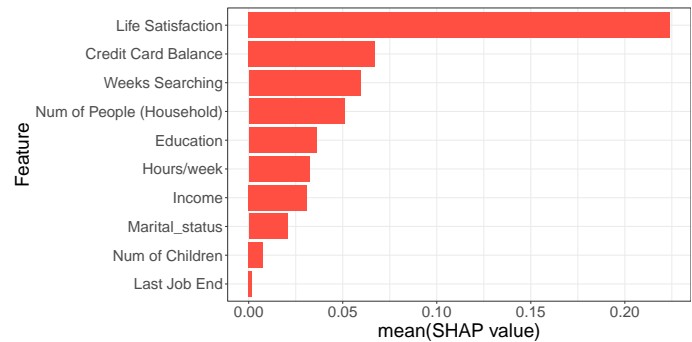

Figure 4: Summary plot of SHAP values for FGBoost applied to emotional well-being data. Features are sorted by their impact in descending order.

the quantile function of $\beta$, respectively. The resulting endpoint of the geodesic $\gamma_{\alpha,\beta}$ is given by $F_\zeta^{-1} = F_\beta^{-1} \circ F_\alpha \circ F_\omega^{-1}$, where $F_\omega^{-1}$ and $F_\zeta^{-1}$ denote the quantile functions of $\omega$ and $\zeta$, respectively.

For the space of networks or symmetric positive-definite matrices equipped with the Frobenius metric, the geodesic corresponds to the line segment connecting the starting and ending points. Assumption 1 is satisfied for Examples 2 and 3 with the geodesic transport map $T_{\gamma_{\alpha,\beta}}(\omega) = \omega + (\beta - \alpha)$.

For the space of compositional data described in Example 4, the geodesic transport map can be interpreted as a rotation of the point $\omega$ along the geodesic determined by $\alpha$ and $\beta$. The tangent vector for the geodesic $\gamma_{\alpha,\beta}$ is given by $v_{\alpha,\beta} = \beta - (\alpha'\beta)\alpha$, whose magnitude and direction encode the geodesic length and directionality needed to move from $\alpha$ toward $\beta$ along the sphere. The geodesic transport map is then defined as:

$$T_{\gamma_{\alpha,\beta}}(\omega) = \text{Exp}_\omega(\theta\frac{v}{\|v\|}) = \cos(\theta)\omega + \sin(\theta)\frac{v}{\|v\|},$$

where $\theta = \arccos(\alpha'\beta)$ is the angle between $\alpha$ and $\beta$, $v = v_{\alpha,\beta} - (\omega'v_{\alpha,\beta})\omega$ is the projection of $v_{\alpha,\beta}$ onto the tangent space at $\omega$, and $\text{Exp}_\omega(\theta v/\|v\|)$ denotes the exponential map at $\omega$ applied to the tangent vector $\theta v/\|v\|$.

This map $T_{\gamma_{\alpha,\beta}}(\omega)$ moves the point $\omega$ along the geodesic connecting it to a new point determined by $\alpha$ and $\beta$, with the direction and distance dictated by the original geodesic $\gamma_{\alpha,\beta}$. The construction ensures that $T_{\gamma_{\alpha,\beta}}(\omega)$ lies on the sphere and preserves the geodesic structure. Assumption 1 is satisfied with this geodesic transport map.

Beyond these examples, extending FGBoost to a new geodesic space requires specifying only two ingredients: the distance function and the associated transport map. For smooth Riemannian manifolds, geodesics and parallel transport are classical and well-studied, with closed-form or numerically stable algorithms widely available. For discrete structures such as trees or networks, geodesics correspond to shortest paths under the chosen metric, and transport can be defined by propagating along these paths. For specialized metrics, such as the BHV metric for phylogenetic trees [5], geodesics and transport maps are explicitly described in the literature.

The transport map assumption, therefore, does not pose a major barrier in practice. For empirical distributions, transport maps are based on empirical quantile functions, which are simple step functions and computationally straightforward. For Riemannian manifolds, parallel transport is standard and efficiently implemented. These principles make FGBoost broadly applicable and provide practitioners with clear guidelines for adapting the method to new domains.

## C Shapley Additive Explanations for Fréchet geodesic boosting

Tree-based regression models, such as boosted trees and random forests, are widely used for their flexibility and ability to model complex non-linear relationships. However, explaining their predictions often receives less attention. Shapley values [55, 37] provide a principled way to measure feature importance for predictive models. Shapley values require retraining the model on all subsets of features $S \subseteq S_{\boldsymbol{x}}$, where $S_{\boldsymbol{x}}$ is the set of all features. They assign an importance value to each feature based on its effect on the model's prediction. To quantify this effect for feature $j$, two models are considered: $f_{S \cup \{j\}}$, trained with feature $j$ included, and $f_S$, trained without feature $j$. The contribution of feature $j$ is measured as the difference in predictions between the two models on the same input, $f_{S \cup \{j\}}(\boldsymbol{x}_{S \cup \{j\}}) - f_S(\boldsymbol{x}_S)$, where $\boldsymbol{x}_S$ represents the values of the input features in the subset $S$. Since the effect of withholding a feature depends on interactions with other features, the differences are computed for all subsets $S \subseteq S_{\boldsymbol{x}} \setminus \{j\}$. The Shapley values are then computed as a weighted average of these differences:

$$\phi_j(f, \boldsymbol{x}) = \sum_{S \subseteq S_{\boldsymbol{x}} \setminus \{j\}} \frac{|S|!\,(|S_{\boldsymbol{x}}| - |S| - 1)!}{|S_{\boldsymbol{x}}|!} \big[ f_{S \cup \{j\}}(\boldsymbol{x}_{S \cup \{j\}}) - f_S(\boldsymbol{x}_S) \big].$$

To extend Shapley values to non-Euclidean outputs, the difference term is replaced by the metric:

$$\phi_j(f, \boldsymbol{x}) = \sum_{S \subseteq S_{\boldsymbol{x}} \setminus \{j\}} \frac{|S|!\,(|S_{\boldsymbol{x}}| - |S| - 1)!}{|S_{\boldsymbol{x}}|!} d\big( f_{S \cup \{j\}}(\boldsymbol{x}_{S \cup \{j\}}), f_S(\boldsymbol{x}_S) \big),$$

where $d$ is the metric of the output space. Because $d(\cdot, \cdot) \geq 0$, these Shapley values quantify the magnitude of each feature's effect and are non-negative by construction.

Shapley Additive Explanations (SHAP) values, introduced by [37], generalize Shapley values to quantify feature contributions for the conditional expectation function of the model's output. Tree SHAP [36], a specialized algorithm for tree-based models, exploits the hierarchical structure of decision trees to efficiently compute SHAP values without retraining the model and evaluating all possible subsets. SHAP value enables both global and local interpretability. Globally, the mean absolute SHAP values highlight the importance of each feature across the dataset, providing insights into which predictors drive the model's behavior. Locally, SHAP values for individual predictions explain how specific feature values contribute to the output. By combining accuracy and transparency, SHAP enhances the interpretability of complex models and builds trust in their predictions. Such an extension can also be useful for other regression models for non-Euclidean outputs.

## D Proof of Proposition 1

*Proof.* To prove that $d_{\mathcal{G}}$ is a valid metric on the space of geodesics $\mathcal{G}(\mathcal{M})$, we verify the following four axioms:

- Identity: For any $\gamma_{\alpha,\beta} \in \mathcal{G}(\mathcal{M})$,

$$d_{\mathcal{G}}(\gamma_{\alpha,\beta}, \gamma_{\alpha,\beta}) = \sqrt{d^2(\alpha, \alpha) + d^2(\beta, \beta)} = 0.$$

- Positivity: For any $\gamma_{\alpha_1,\beta_1}, \gamma_{\alpha_2,\beta_2} \in \mathcal{G}(\mathcal{M})$, if $\alpha_1 \neq \alpha_2$ or $\beta_1 \neq \beta_2$, then $\gamma_{\alpha_1,\beta_1} \neq \gamma_{\alpha_2,\beta_2}$. In this case, at least one of $d^2(\alpha_1, \alpha_2)$ or $d^2(\beta_1, \beta_2)$ is strictly greater than 0, implying

$$d_{\mathcal{G}}(\gamma_{\alpha_1,\beta_1}, \gamma_{\alpha_2,\beta_2}) = \sqrt{d^2(\alpha_1, \alpha_2) + d^2(\beta_1, \beta_2)} > 0.$$

- Symmetry: For any $\gamma_{\alpha_1,\beta_1}, \gamma_{\alpha_2,\beta_2} \in \mathcal{G}(\mathcal{M})$,

$$d_{\mathcal{G}}(\gamma_{\alpha_1,\beta_1}, \gamma_{\alpha_2,\beta_2}) = \sqrt{d^2(\alpha_1, \alpha_2) + d^2(\beta_1, \beta_2)} = d_{\mathcal{G}}(\gamma_{\alpha_2,\beta_2}, \gamma_{\alpha_1,\beta_1}).$$

- Triangle inequality: For any $\gamma_{\alpha_1,\beta_1}, \gamma_{\alpha_2,\beta_2}, \gamma_{\alpha_3,\beta_3} \in \mathcal{G}(\mathcal{M})$, the following holds:

$$
\begin{aligned}
d_{\mathcal{G}}^2(\gamma_{\alpha_1,\beta_1}, \gamma_{\alpha_2,\beta_2}) =& d^2(\alpha_1, \alpha_2) + d^2(\beta_1, \beta_2) \\
\leq& \{d(\alpha_1, \alpha_3) + d(\alpha_2, \alpha_3)\}^2 + \{d(\beta_1, \beta_3) + d(\beta_2, \beta_3)\}^2 \\
=& d^2(\alpha_1, \alpha_3) + d^2(\beta_1, \beta_3) + d^2(\alpha_2, \alpha_3) + d^2(\beta_2, \beta_3) \\
& + 2d(\alpha_1, \alpha_3)d(\alpha_2, \alpha_3) + 2d(\beta_1, \beta_3)d(\beta_2, \beta_3) \\
\leq& d^2(\alpha_1, \alpha_3) + d^2(\beta_1, \beta_3) + d^2(\alpha_2, \alpha_3) + d^2(\beta_2, \beta_3) \\
& + 2\sqrt{\{d^2(\alpha_1, \alpha_3) + d^2(\beta_1, \beta_3)\}\{d^2(\alpha_2, \alpha_3) + d^2(\beta_2, \beta_3)\}} \\
=& \{\sqrt{d^2(\alpha_1, \alpha_3) + d^2(\beta_1, \beta_3)} + \sqrt{d^2(\alpha_2, \alpha_3) + d^2(\beta_2, \beta_3)}\}^2 \\
=& \{d_{\mathcal{G}}(\gamma_{\alpha_1,\beta_1}, \gamma_{\alpha_3,\beta_3}) + d_{\mathcal{G}}(\gamma_{\alpha_2,\beta_2}, \gamma_{\alpha_3,\beta_3})\}^2.
\end{aligned}
$$

Taking the square root on both sides, the triangle inequality follows:

$$
d_{\mathcal{G}}(\gamma_{\alpha_1,\beta_1}, \gamma_{\alpha_2,\beta_2}) \leq d_{\mathcal{G}}(\gamma_{\alpha_1,\beta_1}, \gamma_{\alpha_3,\beta_3}) + d_{\mathcal{G}}(\gamma_{\alpha_2,\beta_2}, \gamma_{\alpha_3,\beta_3}).
$$

Since all four axioms are satisfied, $d_{\mathcal{G}}$ is a valid metric on the space of geodesics $\mathcal{G}(\mathcal{M})$. $\qquad\square$

## E  Proof of Proposition 2

Since $(\mathcal{M}, d)$ is a Hadamard space, the space of geodesic $(\mathcal{G}(\mathcal{M}), d_{\mathcal{G}})$, as a product metric space, is also a Hadamard space. From Proposition 2.3 in [56], for any pair of geodesics $\gamma_0, \gamma_1 \in \mathcal{G}(\mathcal{M})$, there exists a unique geodesic $\Gamma : [0, 1] \mapsto \mathcal{G}(\mathcal{M})$ connecting them, and the intermediate points $\gamma_t = \Gamma(t), t \in [0, 1]$ depend continuously on the endpoints $\gamma_0, \gamma_1$. Furthermore, according to the definition of Hadamard space, for any $\gamma \in \mathcal{G}(\mathcal{M})$

$$
d_{\mathcal{G}}^2(\gamma, \gamma_t) \leq (1 - t)d_{\mathcal{G}}^2(\gamma, \gamma_0) + td_{\mathcal{G}}^2(\gamma, \gamma_1) - t(1 - t)d_{\mathcal{G}}^2(\gamma_0, \gamma_1).
$$

This inequality demonstrates that the function $\psi(\gamma, \cdot)$ is strongly convex over $\mathcal{G}(\mathcal{M})$.

Next, we prove that $\psi(\gamma, \cdot)$ is Lipschitz continuous. Let $\gamma_1, \gamma_2 \in \mathcal{G}(\mathcal{M})$ be two geodesics, it follows that

$$
\begin{aligned}
|\psi(\gamma, \gamma_1) - \psi(\gamma, \gamma_2)| =& \left| d_{\mathcal{G}}^2(\gamma, \gamma_1) - d_{\mathcal{G}}^2(\gamma, \gamma_2) \right| \\
=& \left| d_{\mathcal{G}}(\gamma, \gamma_1) + d_{\mathcal{G}}(\gamma, \gamma_2) \right| \cdot \left| d_{\mathcal{G}}(\gamma, \gamma_1) - d_{\mathcal{G}}(\gamma, \gamma_2) \right| \\
\leq& 2\mathrm{diam}(\mathcal{G}(\mathcal{M})) \cdot \left| d_{\mathcal{G}}(\gamma, \gamma_1) - d_{\mathcal{G}}(\gamma, \gamma_2) \right| \\
\leq& 2\mathrm{diam}(\mathcal{G}(\mathcal{M})) \cdot d_{\mathcal{G}}(\gamma_1, \gamma_2),
\end{aligned}
$$

where $\mathrm{diam}(\mathcal{G}(\mathcal{M}))$ denotes the diameter of $\mathcal{G}(\mathcal{M})$ and is finite since $\mathcal{M}$ is a bounded metric space. Thus $\psi(\gamma, \cdot)$ is Lipschitz continuous with respect to $d_{\mathcal{G}}$.

## F  Proof of Theorem 1

According to Proposition 2, the risk functional $A(\cdot)$ is strongly convex and continuous. Furthermore, $A(\cdot)$ is bounded as $\mathcal{M}$ is bounded. Thus, in particular,

$$
\inf_{F \in \mathrm{span}(\mathcal{F})} A(F) = \inf_{F \in \overline{\mathrm{span}(\mathcal{F})}} A(F),
$$

where $\overline{\mathrm{span}(\mathcal{F})}$ is the closure of $\mathrm{span}(\mathcal{F})$. Since $A(\cdot)$ is strongly convex, there exists a unique function $F^* \in \overline{\mathrm{span}(\mathcal{F})}$ such that

$$
F^* = \arg\min_{F \in \overline{\mathrm{span}(\mathcal{F})}} A(F).
$$

The uniqueness follows directly from the strong convexity of the loss function. Similar arguments apply to the empirical risk functional $A_n(\cdot)$.

## G  Proof of Theorem 2

*Proof.* Define the metric on $\mathrm{span}(\mathcal{F})$ as:

$$d_{\mathcal{F}}(F_1, F_2) = \sup_{\boldsymbol{x} \in \mathbb{R}^p} d_{\mathcal{G}}(F_1(\boldsymbol{x}), F_2(\boldsymbol{x})).$$

It is straightforward to verify that $d_{\mathcal{F}}$ is a valid metric. Specifically, for any $F_1, F_2, F_3 \in \mathrm{span}(\mathcal{F})$,

$$\sup_{\boldsymbol{x} \in \mathbb{R}^p} d_{\mathcal{G}}(F_1(\boldsymbol{x}), F_2(\boldsymbol{x})) \leq \sup_{\boldsymbol{x} \in \mathbb{R}^p} \{d_{\mathcal{G}}(F_1(\boldsymbol{x}), F_3(\boldsymbol{x})) + d_{\mathcal{G}}(F_2(\boldsymbol{x}), F_3(\boldsymbol{x}))\}$$
$$\leq \sup_{\boldsymbol{x} \in \mathbb{R}^p} d_{\mathcal{G}}(F_1(\boldsymbol{x}), F_3(\boldsymbol{x})) + \sup_{\boldsymbol{x} \in \mathbb{R}^p} d_{\mathcal{G}}(F_2(\boldsymbol{x}), F_3(\boldsymbol{x})).$$

Thus, $(\mathrm{span}(\mathcal{F}), d_{\mathcal{F}})$ forms a metric space.

Let $l^{\infty}(\mathrm{span}(\mathcal{F}))$ represent the space of bounded functions on $\mathrm{span}(\mathcal{F})$. To establish that $\sup_{F \in \mathrm{span}(\mathcal{F})} |A_n(F) - A(F)| \to 0$ in probability, it suffices to show that $A_n(\cdot) - A(\cdot)$ weakly converges to 0 in $l^{\infty}(\mathrm{span}(\mathcal{F}))$. Once this weak convergence is shown, Theorem 1.3.6 of [58] can be applied to conclude the result. For a detailed definition of weak convergence in this context, we refer readers to Definition 1.3.3 in [58]. By Theorems 1.5.4 and 1.5.7 of [58], the weak convergence follows upon verifying the following two conditions:

(i) $A_n(F) - A(F) = o_p(1)$ for all $F \in \mathrm{span}(\mathcal{F})$ and

(ii) $A_n(\cdot) - A(\cdot)$ is asymptotically equicontinuous in probability, i.e., for all $\epsilon, \eta > 0$, there exists $\delta > 0$ such that

$$\limsup_n P(\sup_{d_{\mathcal{F}}(F_1, F_2) < \delta} |\{A_n(F_1) - A(F_1)\} - \{A_n(F_2) - A(F_2)\}| > \epsilon) < \eta.$$

To address (i), note that for any $F \in \mathrm{span}(\mathcal{F})$, both $E\{d_{\mathcal{G}}^2(\gamma_{Y_0, Y}, F(\boldsymbol{X}))\}$ and $E\{d_{\mathcal{G}}^4(\gamma_{Y_0, Y}, F(\boldsymbol{X}))\}$ are finite since $\mathcal{M}$ is a bounded metric space. By law of large numbers, $A_n(F) - A(F) = o_p(1)$ for any $F \in \mathrm{span}(\mathcal{F})$.

For (ii), consider any $F_1, F_2 \in \mathrm{span}(\mathcal{F})$, then

$$|\{A_n(F_1) - A(F_1)\} - \{A_n(F_2) - A(F_2)\}|$$
$$\leq |A_n(F_1) - A_n(F_2)| + |A(F_1) - A(F_2)|$$
$$\leq \frac{1}{n} \sum_{i=1}^n |d_{\mathcal{G}}(\gamma_{Y_0, Y_i}, F_1(\boldsymbol{X}_i)) - d_{\mathcal{G}}(\gamma_{Y_0, Y_i}, F_2(\boldsymbol{X}_i))||d_{\mathcal{G}}(\gamma_{Y_0, Y_i}, F_1(\boldsymbol{X}_i)) + d_{\mathcal{G}}(\gamma_{Y_0, Y_i}, F_2(\boldsymbol{X}_i))|$$
$$\quad + |E[\{d_{\mathcal{G}}(\gamma_{Y_0, Y}, F_1(\boldsymbol{X})) - d_{\mathcal{G}}(\gamma_{Y_0, Y}, F_2(\boldsymbol{X}))\}\{d_{\mathcal{G}}(\gamma_{Y_0, Y}, F_1(\boldsymbol{X})) + d_{\mathcal{G}}(\gamma_{Y_0, Y}, F_2(\boldsymbol{X}))\}]|$$
$$\leq 4\{\mathrm{diam}(\mathcal{G}(\mathcal{M}))\}^2 d_{\mathcal{F}}(F_1, F_2)$$
$$= O_p\{d_{\mathcal{F}}(F_1, F_2)\}.$$

Thus,

$$\sup_{d_{\mathcal{F}}(F_1, F_2) < \delta} |\{A_n(F_1) - A(F_1)\} - \{A_n(F_2) - A(F_2)\}| = O_p(\delta),$$

which implies (ii). Finally, by Corollary 3.2.3 in [58] and Theorem 1, it follows that $d_{\mathcal{F}}(F_n^*, F^*) = o_p(1)$. $\square$

## H  Comparison to XGBoost and SketchBoost using vectorized graph Laplacians

To further evaluate the effectiveness of FGBoost, we conducted an additional simulation study comparing it against popular variants of gradient boosting methods, under the same network simulation setup described in Section 5. Since gradient boosting algorithms are not inherently designed to handle metric space-valued responses, we adapted them by vectorizing the graph Laplacians. Specifically, owing to the symmetry and zero row-sum constraints, each Laplacian is fully determined by its strict

Table 5: Average mean squared prediction errors and standard deviations (in parentheses) of Fréchet geodesic boosting (FGBoost), XGBoost and SketchBoost for network outputs.

| $n$ | FGBoost | XGBoost | SketchBoost |
|---|---|---|---|
| 100 | **13.644 (3.140)** | 15.234 (3.319) | 15.391 (2.825) |
| 200 | **10.531 (3.371)** | 11.989 (2.686) | 12.162 (2.421) |
| 500 | **6.912 (1.950)** | 9.035 (2.145) | 8.887 (1.869) |
| 1000 | **5.471 (1.481)** | 7.096 (1.703) | 6.793 (1.443) |

upper triangular entries, which we flattened into a vector-valued output. We considered two baselines: (i) coordinate-wise XGBoost [13], where an independent regressor is trained for each coordinate of the vectorized output, and (ii) SketchBoost [26], a recent multi-output gradient boosting method that jointly models all coordinates. For both methods, default hyperparameters were used. In contrast, FGBoost directly operates on the graph Laplacians as objects in a geodesic metric space.

Table 5 presents average MSPE (with standard deviations) over 500 Monte Carlo replications for varying sample sizes. FGBoost consistently outperforms both XGBoost and SketchBoost across all sample sizes. While SketchBoost improves over XGBoost by leveraging joint modeling, its advantage appears only at larger sample sizes and it remains inferior to FGBoost. These findings highlight a key limitation of vectorization-based approaches: although they enable the application of standard boosting algorithms, they disregard the intrinsic geometry and structural dependencies of graph Laplacians. By directly respecting the non-Euclidean nature of the output space, FGBoost achieves substantial improvements in predictive performance.

## I  Choice of hyperparameters

The hyperparameters for Fréchet geodesic boosting can be selected using a grid search over the candidate values listed in Table 6. The optimal combination of hyperparameters is chosen to minimize the mean squared prediction error for the validation dataset.

Table 6: Hyperparameter settings.

| Learning rate | 0.01 | 0.03 | 0.05 | 0.1 |
|---|---|---|---|---|
| Number of iterations | 50 | 70 | 90 | 100 |
| Depth of each tree | 2 | 3 | 4 | 5 |

## J  Training time and computational complexity

Computational efficiency is a key consideration in the practical deployment of regression methods, particularly in modern applications involving non-Euclidean outputs such as probability distributions and networks. In this section, we provide a systematic comparison of the training times of FGBoost and several state-of-the-art baseline methods across a range of sample sizes. All experiments were conducted on a local machine equipped with an Apple M3 Max chip running macOS Sequoia.

Table 7 reports the training times (in minutes) for sample sizes $n = 100, 200, 500, 1000, 2000$. Among the baseline methods, global Fréchet regression (GFR) is consistently the fastest, as it generalizes linear regression to non-Euclidean settings without introducing significant algorithmic complexity. However, this computational simplicity comes at the cost of substantially reduced model flexibility, since GFR imposes linearity assumptions that may be too restrictive in practice.

Random forest-based methods, Fréchet random forest (FRF) and random forest weighted local linear Fréchet regression (RFWLLFR), are also computationally efficient. Their parallelizable tree-based architectures facilitate fast training, especially on multi-core systems. In contrast, FGBoost trains trees sequentially, leading to a moderately higher computational cost. Nonetheless, this sequential nature allows FGBoost to iteratively correct model bias and effectively capture complex nonlinear relationships, which is particularly advantageous when modeling outputs in curved or high-variance metric spaces.

Table 7: Training time in minutes across different sample sizes.

| $n$ | FGBoost | GFR | SDR | IFR | FRF | RFWLLFR |
|---|---|---|---|---|---|---|
| 100 | 0.54 | 0.003 | 1.27 | 2.24 | 0.24 | 0.24 |
| 200 | 0.95 | 0.006 | 3.89 | 7.00 | 0.55 | 0.55 |
| 500 | 2.41 | 0.020 | 19.68 | 25.65 | 1.56 | 1.55 |
| 1000 | 3.93 | 0.080 | 71.45 | 137.82 | 3.41 | 3.37 |
| 2000 | 7.14 | 0.270 | — | — | 6.75 | 6.77 |

Table 8: Average mean squared prediction errors and standard deviations (in parentheses) of Fréchet geodesic boosting (FGBoost), global Fréchet regression (GFR) [47], sufficient dimension reduction (SDR) [62], single index Fréchet regression (IFR) [4], Fréchet random forest (FRF) [10] and random forest weighted local linear Fréchet regression (RFWLLFR) [49] for National Health and Nutrition Examination Survey data.

| FGBoost | GFR | SDR | IFR | FRF | RFWLLFR |
|---|---|---|---|---|---|
| **0.054 (0.001)** | 0.059 (0.001) | 0.065 (0.006) | 0.071 (0.012) | 0.058 (0.001) | 0.073 (0.003) |

Methods based on dimension reduction, including sufficient dimension reduction (SDR) and single index Fréchet regression (IFR), are substantially more computationally intensive. These methods involve iterative estimation of latent structures and repeated geodesic evaluations, resulting in poor scalability. At $n = 2000$, the computational cost of SDR and IFR became prohibitive, and we were unable to obtain results within a reasonable time frame.

Overall, FGBoost achieves a favorable trade-off between computational cost and modeling flexibility. While it is not the fastest method in absolute terms, its ability to scale to large datasets and to accommodate complex, non-Euclidean output structures makes it a competitive and practical choice in modern regression settings.

## K  Additional real-world data application: National Health and Nutrition Examination Survey

To further assess the empirical performance of FGBoost, we analyzed a fourth real-world dataset from the National Health and Nutrition Examination Survey (NHANES) 2005–2006. NHANES is a large-scale survey that evaluates the health and nutritional status of U.S. adults and children through interviews, physical examinations, and laboratory tests. In this cycle, participants aged six years and older were asked to wear an ActiGraph 7164 accelerometer on the right hip for seven consecutive days. The device recorded physical activity intensity in counts per minute (CPM) at 1-minute resolution, beginning at 12:01 am on the day following the health examination and removed only for sleep, swimming, or bathing. These accelerometer data have been widely used to study the relationship between physical activity and health outcomes [31, 24].

We focused on modeling the distribution of physical activity intensity as a non-Euclidean response, using demographic and health-related variables as predictors. For each participant, activity values equal to zero or exceeding 1000 CPM were excluded, since zeros may correspond to various low-activity states (e.g., sleep or device non-wear) and values above 1000 CPM are typically considered measurement artifacts. The remaining activity counts over the seven days were concatenated to form the empirical distribution of each participant's activity intensity. Similar distributional representations have been employed in recent studies [11, 34, 39]. The predictor set comprised 13 demographic and anthropometric variables: gender, age, race/ethnicity, veteran status, education (college or above), household income ($\leq 35,000$), marital status, weight, height, body mass index (BMI), thigh circumference, waist circumference, and upper arm length. To ensure data quality and reliable coverage, we selected the 200 participants with the most valid observations and performed 10-fold cross-validation over 20 runs for model evaluation.

Table 8 reports the AMSPE for FGBoost and competing regression methods. FGBoost achieves the best predictive accuracy, outperforming all alternatives. These results demonstrate FGBoost's ability

to capture complex regression relationships when the outcome is an empirical distribution derived from high-frequency sensor data.

# L    Limitations

While FGBoost provides a flexible framework for regression with metric space-valued outputs, it has several limitations. First, although boosting reduces bias and can help control variance, it is still susceptible to overfitting, particularly when the base learners are overly complex or the number of boosting rounds is large. In FGBoost, we address this by limiting tree depth and applying early stopping, but the risk remains, especially in small-sample or high-noise settings.

Second, our theoretical analysis depends on the assumption that the output space is a Hadamard space, a condition met by many practical metric spaces, but nonetheless restrictive. Broadening the analysis to encompass more general geodesic metric spaces would improve the generality of the theoretical guarantees. From an implementation perspective, however, FGBoost can be applied in any geodesic space. For example, our experiments on compositional data (Appendix A) involve the positive hypersphere, which is not a Hadamard space, and demonstrate strong empirical performance. This suggests that the method is practically robust beyond the confines of the Hadamard assumption, even though formal guarantees do not yet extend to these cases.

Third, while Section 4 establishes strong convexity of the risk functional, we do not provide a formal convergence proof for FGBoost. Classical convergence analyses for boosting rely on Banach space structures, where linear operations enable the use of Taylor expansions or Gâteaux derivatives [63]. In geodesic metric spaces, such tools are unavailable, and new geometric techniques would be required to establish descent guarantees. Developing these tools is an important avenue for future research.

Fourth, FGBoost is inherently more computationally intensive than scalar-based boosting methods due to the need for metric evaluations and Fréchet mean computations at each iteration. While many commonly used spaces admit closed-form solutions (e.g., Wasserstein distributions, SPD matrices with power metrics, networks with Frobenius metric) or efficient iterative algorithms (e.g., proximal point methods) [1], these steps still add overhead compared to simple arithmetic operations. Our implementation adopts a modular design that separates metric-specific primitives from the core boosting loop, allowing extensibility across different geodesic spaces. Nevertheless, developing a fully optimized and unified library that achieves the efficiency of established boosting frameworks such as XGBoost or LightGBM is an ambitious but promising avenue for future work.

Finally, while we extend SHAP values to interpret FGBoost predictions, the model's ensemble structure, built from numerous weak learners, makes it inherently difficult to interpret. This limits transparency and may pose challenges in domains where understanding the model's decision process is essential.

Future work could explore additional regularization strategies, such as penalizing leaf weights or incorporating dropout-like mechanisms, aiming to enhance robustness.

## M   Additional tables

Table 9: Predictors of human mortality data.

| Category | Variables | Explanation |
|---|---|---|
| Demography | 1. Population Density | population per square kilometer |
| | 2. Sex Ratio | number of males per 100 females in the population |
| | 3. Mean Childbearing Age | average age of mothers at the birth of their children |
| Economics | 4. GDP | gross domestic product per capita |
| | 5. GVA by Agriculture | percentage of agriculture, hunting, forestry, and fishing activities of gross value added |
| | 6. CPI | consumer price index treating 2010 as the base year |
| | 7. Unemployment Rate | percentage of unemployed people in the labor force |
| | 8. Health Expenditure | percentage of expenditure on health of GDP |
| Environment | 9. Arable Land | percentage of total land area |

Table 10: Predictors of New York City taxi network data.

| Category | Variables | Explanation |
|---|---|---|
| Weather | 1. Temp | daily average temperature |
| | 2. Humidity | daily average humidity |
| | 3. Wind | daily average windspeed |
| | 4. Pressure | daily average barometric pressure |
| | 5. Precipitation | daily total precipitation |
| Day | 6. Mon to Thur | indicator for Monday to Thursday |
| | 7. Friday or Saturday | indicator for Friday or Saturday |
| Trip | 8. Passenger Count | daily average number of passengers |
| | 9. Trip Distance | daily average trip distance |
| | 10. Fare Amount | daily average fare amount |
| | 11. Tip Amount | daily average tip amount |
| | 12. Tolls Amount | daily average tolls amount |

