# OpenReview forum: "Fréchet Geodesic Boosting"
_NeurIPS.cc/2025/Conference — NeurIPS 2025 poster_

### Official Review · Reviewer_rLbt · 2025-07-01

**Clarity:** 3
**Significance:** 3
**Originality:** 3
**Rating:** 4
**Confidence:** 5

**Summary:**

The paper Fréchet Geodesic Boosting generalizes gradient boosting to Riemannian manifolds by introducing a novel algorithm Fréchet Geodesic Boosting (FGBoost). Traditional boosting methods operate in Euclidean spaces, where residuals are computed as vector differences. However, in non-Euclidean settings, subtraction is not well-defined. FGBoosting replaces residuals with tangent vectors derived from the Fréchet mean, allowing the model to minimize a loss defined over geodesic distances.

**Questions:**

1. What assumptions are required for the manifold to ensure the algorithm is well defined? Like, completeness, injectivity radius, etc.
2. How does the algorithm ensure consistency across tangent spaces at different base points during updates?

**Ethical Concerns:**

["NO or VERY MINOR ethics concerns only"]

**Final Justification:**

The rebuttal is well explained and the authors solved the most of my concerns.

**Limitations:**

Yes

**Paper Formatting Concerns:**

No formatting issues

**Quality:**

3

**Strengths And Weaknesses:**

Strengths:
1. The paper clearly identifies the limitations of classical boosting in curve spaces and provides a nice solution.
2. This is a novel step in extending boosting to settings with non-Euclidean geometry.

Weaknesses:
1. It would be better a further theoretical discussion. The paper doesn't provide formal convergence guarantees.
2. The algorithm relies on having nice exponential and logs maps, which can be a limitation on certain manifolds.
3. Learning tangent vectors using regressors may introduce approximation errors, especially in curved regions where tangent space linearization is less accurate.
FGBoost is a powerful method for learning on Riemannian manifolds and significantly extends boosting to curved domains but the results present computational challenges, limited theoretical analysis, and dependence on manifold-specific operations.

---

> ### Author Rebuttal · Authors · 2025-07-28
>
> We thank you for your time and for providing detailed feedback on our work. We are encouraged that you recognized the novelty of extending boosting to settings with non-Euclidean geometry and appreciated our solution to the limitations of classical boosting.
>
> We would like to begin by clarifying a central point about our paper's scope, as we believe it may resolve many of the concerns you raised. Our framework, FGBoost, is developed for the broad class of **geodesic metric spaces** (Bridson & Haefliger, 2013), which is a much more general setting than smooth Riemannian manifolds.
>
> While smooth Riemannian manifolds are an important example of geodesic spaces, our methodology does not require the additional structure of a manifold, and specifically does not in any way use tangent spaces or Riemannian exponential and log maps. Our algorithm operates intrinsically in a geodesic metric space using only the metric and the geodesics (shortest paths) it induces. This is why our "pseudo-residuals" are defined as geodesics themselves. We specifically note that they are not tangent vectors. This general formulation is a key strength, allowing FGBoost to be applicable for a wide range of data.
>
> We only mention tangent planes and tangent vectors in Appendix A since in this special case the data are located on a sphere and we use Riemannian maps and tangent spaces in this special case to generate random data for the simulation but not for our FGBoost approach, which does not utilize these tools as they are generally not available. Indeed, for our primary experiments in Section 5 of the main paper, we use only the geodesic structure of the spaces and no tools that are specific to Riemannian manifolds.
>
> We hope this clarification helps address your specific points below.
>
> ---
>
> > Reliance on exponential/log maps and tangent spaces
>
> Your concerns about requiring "nice exponential and log maps" and errors from "tangent space linearization" would be valid points if our approach were using Riemannian tools. However, as explained above, this is not the case. As mentioned, since our method is formulated for general geodesic spaces, it does not use tangent vectors, exponential maps, or log maps. All operations are defined directly in the original metric space using geodesics.
>
> ---
>
> > Required assumptions (completeness, injectivity radius, etc.)
>
> Because we do not assume a manifold structure, our assumptions are on the metric space itself. Our primary theoretical assumption, stated in Section 4, is that the output space is a Hadamard space (a complete geodesic space with non-positive curvature, which guarantees the existence and uniqueness of the geodesics and Fréchet means). This assumption is only required for the supporting theory but not for the implementation of the method. Since there is no assumed manifold structure, there are no assumptions on completeness or injectivity radius in the context of a manifold.
>
> ---
>
> > Consistency across tangent spaces
>
> As our method does not operate in tangent spaces, the issue of ensuring consistency between them does not arise. All updates are performed intrinsically within the original metric space via the defined geodesic operations.
>
> ---
>
> > Convergence proof
>
> This is a good point. As we mentioned in the response to **Reviewer XGWn**, classical convergence proofs for boosting rely on tools like Taylor expansions or functional derivatives, which are not available in our general geodesic setting. Developing the geometric tools for the proof in general geodesic metric spaces (especially spaces that are not Hadamard spaces) is a challenging but important direction for future work. We will add a discussion of this limitation to the paper.
>
> ---
>
> We thank you again for your time and feedback. We hope our response has clarified the central scope of our work, focusing on **general geodesic metric spaces** rather than the more restricted case of smooth Riemannian manifolds, and in doing so, addressed the technical concerns you raised. We believe this **generality** is a key strength of our contribution. In the revised manuscript, we will make this distinction even clearer in the introduction and methodology to prevent future misinterpretation. We would be grateful for your feedback on whether these clarifications have provided a clearer assessment of our contributions.
>
> ---
>
> **References**
> - Bridson, M.R. and Haefliger, A., 2013. *Metric Spaces of Non-Positive Curvature* (Vol. 319). Springer Science & Business Media.

---

> > ### Comment · Reviewer_rLbt · 2025-08-05
> > **Clarification**
> >
> > Thank you for the detailed clarification. I will rise my score.

---

> > > ### Author Response · Authors · 2025-08-05
> > >
> > > Thank you for your follow-up and for reconsidering your assessment after our clarifications. We greatly appreciate your engagement and will incorporate the proposed revisions to further improve clarity and accessibility in the final version.

---

### Official Review · Reviewer_XGWn · 2025-07-02

**Clarity:** 3
**Significance:** 4
**Originality:** 4
**Rating:** 5
**Confidence:** 4

**Summary:**

The authors propose Fréchet Geodesic Boosting (FGBoost), a boosting method that works with outputs residing in unique geodesic spaces.
Some brief theoretical analysis is conducted, establishing, Under the additional assumption of the geodesic space being a Hadamard space, that the underlying optimization is well-posed and a uniform convergence bound (showing that the minimizer of the empirical risk functional converges to that of the minimizer of the populational risk functional).
A significant amount of numerical experiments are also presented, showcasing the superiority of the proposed FGBoost algorithm throughout.
The authors also generalize SHAP explanations to their models (a standard tool in practical applications of existing boosting models) in the supplementary material.

**Questions:**

I'd appreciate some clarifications on the points raised in the 'Strengths&Weaknesses' text:
- How is each iteration's argmin solved?
- Any blockers on a convergence proof?
- Is working with the MSE-like objective fundamental to your approach?
- And the clarifications on the proof of Proposition 2.

**Ethical Concerns:**

["NO or VERY MINOR ethics concerns only"]

**Final Justification:**

My main concerns (explaining how the tree fit is done, convergence guarantees and limited flexibility wrt loss) were appropriately clarified in the rebuttal phase, and corresponding improvements to the writing of the paper were promised. I believe both the problem and proposed solution of the paper are of interest to the NeurIPS community, and thus recommend acceptance.

**Limitations:**

Limitations were appropriately discussed. (Except perhaps for a minor boundedness assumption that could be more explicitly stated, but which seems fairly mild.)

**Paper Formatting Concerns:**

None.

**Quality:**

3

**Strengths And Weaknesses:**

The problem of boosting with non-Euclidean outputs is a very relevant one, and the focus on geodesic spaces is quite sensible.
I find the work to be fairly easy to read, interesting and with convincing results.
I am leaning towards acceptance, and just have a couple of points in which I would appreciate some clarification on.

- On the step-wise fitting procedure: in lines 182-186, it is said that $\hat{f}_k$ is chosen as the minimizer
  $$ \hat{f}\_k = \arg \min\_f \frac{1}{n} \sum\_{i=1}^n d^2 ( Y\_i, T\_{\hat{F}\_{k-1}(\mathbf{X}_i) \oplus \\{\nu \odot f(\mathbf{X}\_i)\\}}(Y\_0) ). $$
  However, it is not 100% clear how this argmin is solved in practice; this is not a simple MSE minimization problem, which e.g. CART solves for trees; the outputs of $f$ gets added to the current model and then passes through the transport map $T$, to only then be piped through a (possibly non-MSE?) squared metric.
  Overall it looks like quite a fairly tricky optimization problem, somewhat similar to those posed in the combinatorial boosting literature (e.g. [ExactBoost](https://proceedings.mlr.press/v151/csillag22a.html) and [IPBoost](https://proceedings.mlr.press/v119/pfetsch20a.html); in fact, it seems particularly close to Equation (4) in ExactBoost, which requires meta-heuristic approaches to be solved).
  On the other hand, the authors say that they use tree fitting for FGBoost, so it feels as though I am missing something.

- In the theoretical analysis, I felt the lack of a proof of convergence: FGBoost is trained in a greedy fashion, and so is (in principle) a descent algorithm. In Section 4 it is proven that the risk functional is strongly convex (and thus has a unique minimizer), but this need not imply that FGBoost will converge to it. While not critical, this would be a nice result to have. Is there some particular difficulty in proving such a result for FGBoost?

- Other loss functions: one of the great boons of gradient boosting is that one can easily change the loss function being fitted without having to tweak the base learners. FGBoost, in comparison, seems hard-wired to an MSE-like objective based on the space's metric. Is this fundamental to the approach?

- Minor clarifications on the proof of Proposition 2:
    - It's not immediately clear why the first display equation (line 573-574 upper bound on $d_{\mathcal{G}}^2 (\gamma, \gamma_t)$) holds; I presume it is due to the Hadamard space assumption?
    - Why must the intermediate points $\gamma_t$ (line 573) depend continuously on the endpoints $\gamma_0, \gamma_1$?
    - Also, this proof invokes the diameter of $\mathcal{G}(\mathcal{M})$ (as does Theorem 2), which is stated to be finite because $\mathcal{M}$ is bounded; however, I could not find this assumption stated elsewhere. (The assumption is fine, just needs to be stated more explicitly in the main text.)

---

> ### Author Rebuttal · Authors · 2025-07-28
>
> We sincerely thank you for your positive assessment and detailed, constructive feedback. We are encouraged that you found the problem "**very relevant**," the work "**fairly easy to read, interesting and with convincing results**," and the focus on geodesic spaces "**quite sensible**."
>
> We appreciate the opportunity to clarify the points you raised regarding the practical optimization, the convergence proof, and the choice of loss function. Below, we address each of your weaknesses and questions.
>
> ---
>
> > Practical implementation of the `argmin` step for $\hat{f}_k$
>
> Thank you for this insightful comment. We will use your excellent feedback to improve the manuscript's clarity. You are correct in observing that the objective function for the base learner $\hat{f}_k$, as written in the paper,
>
> $\hat{f}_{k}=\underset{f}{\operatorname{argmin}}\frac{1}{n}\sum\_{i=1}^nd^2(Y_i, T\_{\hat{F}\_{k-1}(\mathbf{X}_i)\oplus\{\nu\odot f(\mathbf{X}_i)\}}(Y_0))$
>
> is a complex, nested optimization that is not directly solved by standard tree-fitting algorithms.
>
> The key detail, which we will make more explicit in our revision, is that we employ a practical, greedy approximation common to the gradient boosting framework. The procedure is a two-stage process:
> - **Compute pseudo-residuals**: At each iteration $k$, we first compute the target geodesics connecting the current prediction to the actual observation, $\gamma_{\hat{Y}\_{i}^{k-1}, Y_{i}}$. These geodesics act as pseudo-residuals for the current fitting stage.
> - **Fit a tree to pseudo-residuals**: We then train a standard decision tree on the data pairs $\{(X_{i}, \gamma_{\hat{Y}\_{i}^{k-1}, Y_{i}})\}_{i=1}^{n}$. For each terminal leaf, the prediction is the Fréchet mean of the pseudo-residuals within that leaf. As we note (lines 192-194), this calculation is computationally feasible because the optimization decouples into two independent components.
>
> This also clarifies the distinction of our proposed method from the combinatorial boosting methods like `ExactBoost` and `IPBoost` that you mentioned. In our FGBoost framework, the base learner is trained on a **pre-computed target**, the pseudo-residual geodesics. This differs from combinatorial methods where the base learner itself must often solve a more complex optimization problem (e.g., finding an optimal subset or partition) rather than fitting to a simple, pre-defined target.
>
> ---
>
> > Convergence guarantee
>
> This is an excellent point. The primary difficulty lies in the geometric setting. Classical convergence proofs for boosting (e.g., Zhang & Yu, 2005) rely on tools like **Taylor expansions** or **Gâteaux derivatives** to analyze the **functional descent** at each step. These techniques require at least a Banach space structure, where linear operations like addition, subtraction, and scalar multiplication are well-defined. In contrast, our setting involves general geodesic metric spaces, where such operations are not available. As a result, these analytical tools cannot be directly applied without imposing much stronger assumptions. We believe that developing the geometric tools necessary for such a proof is a significant undertaking and an exciting direction for future research. We will add a discussion of this limitation and future direction to the paper.
>
> ---
>
> > Flexibility of the loss function
>
> You raise an excellent and fundamental point about the flexibility of the loss function, which is indeed a hallmark of the standard gradient boosting framework. Our approach is best understood as a direct geometric generalization of the most common boosting setup: using squared error (MSE) loss for regression.
>
> In that classic Euclidean setting, the negative gradient of the loss function simplifies to the residual ($Y_i - F_{k-1}(X_i)$). The base learner, typically a tree, is then tasked with fitting these residuals.
>
> For outputs in a geodesic metric space, we propose that the most natural and principled generalization of the "**residual**" is the **geodesic** path connecting the current prediction to the true observation ($\gamma_{\hat{Y}\_{i}^{k-1}, Y_{i}}$). Our use of the squared-metric loss is directly motivated by this analogy; it allows the target of the base learner to be this generalized residual, which is the direct counterpart to the negative gradient in the MSE case.
>
> Conversely, other common regression losses, such as Huber or log-loss, are **not well-defined** for outputs in a general metric space as they rely on algebraic structures (e.g., subtraction, addition) not present in these spaces. Defining meaningful and well-behaved counterparts for these other losses in a geometric context is a significant research challenge in its own right. Therefore, our choice is a consequence of providing a direct and geometrically sound generalization of the canonical gradient boosting algorithm. We will revise the manuscript to make this motivation clearer.
>
> ---
>
> > Minor clarifications on the proof of Proposition 2
>
> We thank you for carefully reading the proofs and for these helpful questions.
> - **First display equation in Prop 2 (line 573-574).** Your presumption is exactly right. This inequality is a defining property of convexity in Hadamard spaces and follows directly from the assumption that the space is Hadamard. We will add a more explicit reference to make this clearer.
> - **Continuous dependence of intermediate points (line 573).** This is also a standard property of geodesics in Hadamard spaces; see Proposition 2.3 of Sturm (2003). The map that defines the geodesic path is continuous with respect to its endpoints. We will clarify this in the proof.
> - **The boundedness assumption.** Thank you for noting this. The assumption is stated in the main text in Section 2, line 78: "Let $(\mathcal{M}, d)$ be a bounded metric space.". However, we agree that its importance for the proofs in the appendix could be made more prominent. We will reiterate this assumption in Section 4 before the theoretical results to improve clarity.
>
> ---
>
> We sincerely value the time you've dedicated to reviewing our work and deeply appreciate your insightful questions, which have enhanced the quality of our paper. Please let us know if our response has sufficiently resolved your concerns, and we are more than happy to take any additional questions.
>
> ---
>
> **References**
> - Sturm, K.T., 2003. Probability measures on metric spaces of nonpositive curvature. *Heat Kernels and Analysis on Manifolds, Graphs, and Metric Spaces (Paris, 2002). Contemp. Math., 338. Amer. Math. Soc., Providence, RI*, 338, pp. 357–390.
> - Zhang, T. and Yu, B., 2005. Boosting with early stopping: Convergence and consistency. *Annals of Statistics*, 119(548), pp. 1538-1579.

---

> ### Comment · Reviewer_XGWn · 2025-08-02
>
> Thank you for your response; it has appropriately answered my concerns, and so I will be raising my score accordingly.
>
> I strongly suggest the authors incorporate the changes they propose.

---

> > ### Author Response · Authors · 2025-08-02
> >
> > Thank you for your positive feedback and your willingness to reassess the score after our clarifications. We greatly appreciate your constructive suggestions, which we will incorporate into the revised manuscript to further improve clarity and accessibility.

---

### Official Review · Reviewer_aXWy · 2025-07-03

**Clarity:** 4
**Significance:** 3
**Originality:** 3
**Rating:** 5
**Confidence:** 4

**Summary:**

This paper introduces Fréchet Geodesic Boosting (FGBoost), a novel method that extends the gradient boosting framework to regression settings with Euclidean predictors and non-Euclidean outputs residing in geodesic metric spaces. The key idea is to use geodesics—the shortest paths between points in these spaces—as a proxy for residuals, which are not well-defined in non-Euclidean domains. This viewpoint allows a smooth extension of the gradient boosting framework to non-Euclidean settings. The author(s) establish theoretical guarantees for FGBoost, including the existence and uniqueness of the solution and the consistency of the risk minimizer, under the assumption that the output space is a Hadamard space. The author(s) also provide SHAP values for interpretability of FGBoost. The method's improved performance against competitors is demonstrated through three simulation studies and three real-world datasets.

**Questions:**

I alluded to some of these questions in my strengths and weaknesses discussion:

- The theoretical justification of the methodology seems grounded in Hadamard spaces. The paper does well in explaining the three settings presented. However, I wonder how robust the results are to metrics that are not those listed. Perhaps an experiment with a metric that does not lead to a Hadamard space could be helpful to see how robust this approach could be. A response here would clarify the practical boundaries of the method.

- The general framework for geodesic metric spaces is great, but it seems a practitioner would have to derive the geodesic transport map themselves if their problem isn't one of the three examples in the paper. This might limit how widely the method is adopted. Perhaps more guidance on the principles for constructing these maps could be helpful for new users. Addressing this could make the work much more accessible.

- The implementation of tree-based FGBoost relies on finding the Fréchet mean of residual geodesics in each leaf. In the analysis, the chosen metric for the space of geodesics, $d_\mathcal{G}$ conveniently decouples the metric to align with the empirical loss. Does this decoupling on the geodesic space have any tradeoffs among computability/convenience and geometry of geodesics space? May not be necessary, but a comment or two could help clarify here.

**Ethical Concerns:**

["NO or VERY MINOR ethics concerns only"]

**Final Justification:**

I thought the initial paper was strong, and the authors were able to address my questions. Their overall discussions with other reviewers were also very thorough.

**Limitations:**

Yes.

**Paper Formatting Concerns:**

Headings should be lowercase aside from the first letter. There are inconsistencies in the section headings.

**Quality:**

4

**Strengths And Weaknesses:**

**Quality:**

I believe this paper is of high quality. The presentation of the method and its theoretical and experimental justifications are strong. The evaluation across three different non-Euclidean output types, plus several real-world datasets, makes the demonstrated performance advantages convincing.

**Clarity:**

The paper is very clear and well-written. The main contributions are highlighted and previous work in regression in non-Euclidean settings is presented. The development of FGBoost is smooth and well-motivated, drawing a clear link that FGBoost is a generalization of standard gradient boosting methods for geodesic metric spaces. Both theoretical and experimental justifications are convincing, and the three clear examples are helpful. I also appreciate the discussions on scalability and limitations in the Appendix, though it would help with transparency if a reference to this was added in the main text.

**Significance + Originality:**

I’m not very familiar with the related literature, but it seems that the work is novel in tackling general regression problems with Euclidean predictors and non-Euclidean outputs using a boosting framework. However, the paper references Matsubara (2024), which has a very similar motivation and core insight, leveraging Wasserstein gradients in the space of distributions. This could diminish the novelty a bit, as a key idea of using a geometric proxy for residuals is captured in that prior work. Still, this paper's contribution is distinct in its generality beyond distributions and its theoretical guarantees.

I think a couple points may limit the method's scope of impact.

- While placing FGBoost in the general context of geodesic metric spaces is a strength, for actual usage, it seems a practitioner would have to derive the geodesic transport map if their problem is not one of the three examples provided. This may limit the adoptability of the method, and more guidance here could be helpful.
- It is also not clear how other metrics might impact FGBoost, especially those that result in non-Hadamard spaces where the theoretical guarantees do not hold. An experiment exploring this could possibly strengthen the paper by defining the practical boundaries of the algorithm.

[1] Takuo Matsubara. Wasserstein gradient boosting: A framework for distribution-valued supervised learning. In *Advances in Neural Information Processing Systems*, 2024.

---

> ### Author Rebuttal · Authors · 2025-07-28
>
> We are very grateful for your thoughtful and encouraging review. We are delighted that you found the paper to be of "**high quality**" and "**very clear and well-written**," with a "**smooth and well-motivated**" development of the FGBoost framework. We especially appreciate you noting the strength of our evaluation across multiple data types and the helpfulness of our examples.
>
> Your feedback is invaluable, and we will use it to further improve the paper. Below, we address your comments on novelty and scope, as well as your specific questions.
>
> ---
>
> > Relation to Matsubara (2024) and novelty
>
> We appreciate your observation regarding Matsubara (2024) and welcome the opportunity to clarify the key differences. Although both works share a geometric motivation, the problem formulations and objectives are fundamentally different.
>
> Learning target:
> - FGBoost generalizes regression by learning a map from predictors to **outputs in a general geodesic space**. The prediction target is the observed non-Euclidean object itself.
> - WGBoost (Matsubara, 2024) focuses on evidential learning and uncertainty quantification for **scalar-output problems**. It constructs an intermediate target for each observation, the posterior distribution, using Wasserstein geometry (see Example 1, Sections 4.1 and 4.2 in Matsubara, 2024).
>
> Algorithmic and theoretical differences:
> - **Generality**: FGBoost applies broadly to geodesic spaces (e.g., univariate distributions, networks, SPD matrices, compositional data), whereas WGBoost is tailored to uncertainty quantification in scalar regression.
> - **Theory**: We establish risk convexity and the convergence of the empirical risk functional under Hadamard space assumptions, while WGBoost primarily offers methodological and empirical contributions.
>
> We will update the Related Work section to make these distinctions explicit. Thank you for prompting this clarification.
>
> ---
>
> > Practicality of geodesic transport maps
>
> Thank you for highlighting this important point on accessibility and ease of adoption. While our paper illustrates FGBoost on four representative settings including univariate distributions, networks, SPD matrices, and compositional data (Appendix A), we agree that offering general guidance for practitioners will improve clarity.
>
> Our framework is inherently modular and unified: extending FGBoost to a new geodesic space primarily requires specifying (i) the distance function $d$ and (ii) the geodesic transport map. Both components are standard and easily accessible for most geodesic spaces of interest.
> - **Riemannian manifolds**: For smooth Riemannian manifolds, geodesics and their associated transport maps (e.g., parallel transport) are classical and generally well-studied, with closed-form or numerically stable solutions.
> - **Other spaces**: For discrete structures such as trees, geodesics are typically defined via shortest paths, which can be computed efficiently using standard algorithms (e.g., Dijkstra), or are explicitly available for specified metrics such as the BHV metric (Billera et al., 2001).
>
> We will make these principles explicit in the revision and provide additional guidance in the Appendix to support practical adoption.
>
> ---
>
> > Robustness beyond Hadamard spaces
>
> You correctly note that our theoretical guarantees require the assumption that the underlying geodesic space is a Hadamard space. We would like to clarify that this assumption is primarily for the **theoretical analysis** (e.g., proving strong convexity of the loss and uniqueness of the solution). From an **implementation perspective**, FGBoost is applicable for any geodesic space. Our experiments on **compositional data**, presented in Appendix A, provide a practical demonstration of FGBoost in a **non-Hadamard space** (the positive hypersphere) and demonstrate good performance. This showcases the practical robustness of our method beyond the confines of the Hadamard space assumption. We will add a sentence to the main text to highlight this further.
>
> ---
>
> > Metric on the space of geodesics
>
> The choice of $d_\mathcal{G}$ was indeed motivated by its convenient property of decoupling the Fréchet mean calculation into two independent, tractable problems. This makes the tree-based implementation efficient.
>
> The important thing to note is that this choice does not introduce trade-offs or limitations related to the geometry of the underlying output space $\mathcal{M}$. The metric $d_\mathcal{G}$ is on the space of geodesics $\mathcal{G}(\mathcal{M})$, not on $\mathcal{M}$ itself. The core of our algorithm, the pseudo-residuals and aggregation, operates directly on $\mathcal{M}$ using its intrinsic metric $d$. The decoupling is purely a computational convenience for the leaf node estimation that does not compromise the geometric integrity of the approach. We will add a brief comment to this effect to clarify that this choice is for computability without geometric sacrifice.
>
> ---
>
> Finally, we thank you for the formatting suggestion. We will correct the section heading capitalization in the final version. Your detailed feedback will significantly strengthen our paper, and we are very grateful. Please let us know if our response has sufficiently resolved your concerns, and we are more than happy to take any additional questions.
>
> ---
>
> **References**
> - Billera, L.J., Holmes, S.P. and Vogtmann, K., 2001. Geometry of the space of phylogenetic trees. *Advances in Applied Mathematics*, 27(4), pp.733-767.
> - Matsubara, T., 2024. Wasserstein gradient boosting: A framework for distribution-valued supervised learning. *Advances in Neural Information Processing Systems*, 37, pp.118994-119024.

---

> > ### Comment · Reviewer_aXWy · 2025-08-05
> >
> > Thank you for your detailed response! I appreciate you taking the time to address my concerns and questions, and I will maintain my score.

---

> > > ### Author Response · Authors · 2025-08-05
> > >
> > > Thank you for your thoughtful feedback and for reviewing our clarifications. We appreciate your engagement and are glad we could address your concerns. We will incorporate the proposed revisions to further improve clarity and accessibility in the final version.

---

### Official Review · Reviewer_HbNk · 2025-07-10

**Clarity:** 4
**Significance:** 3
**Originality:** 4
**Rating:** 5
**Confidence:** 4

**Summary:**

The paper proposes a new gradient boosting framework which adapts for non-Euclidean output spaces. The new framework starts from a Frechet mean of regression labels, and adds geodesics in each boosting round to the current prediction of each sample. FGBoost uses decision trees as base learners to partition the data into regions. Optimal geodesics to add are calculated for samples in each region as the Frechet mean of the residual geodesics of these samples. When the output space is a Hadamard space, there exists a consistent esitmator to the global unique minimizer of the boosting loss.

Numerical experiments are conducted in both simulation datasets and real-world applications, covering probability and graph outputs. FGBoost shows superior performance over existing regression models for non-Euclidean outputs.

**Questions:**

1. Why are all the SHAP values shown in Figure 2 positive? Are the absolute values plotted? Or just the SHAP values of features with highest positive values are plotted?
2. How the trees are constructed in FGBoost?
3. In the experiment comparing with XGBoost, XGBoost is trained to fit each value of the Laplacian matrix. It would be interesting to see how the performance changes with traditional gradient boosting when fitting multiple values together, given the recent progress of multiple output gradient boosting decision trees (e.g., https://proceedings.neurips.cc/paper_files/paper/2022/file/a36c3dbe676fa8445715a31a90c66ab3-Paper-Conference.pdf).

**Ethical Concerns:**

["NO or VERY MINOR ethics concerns only"]

**Final Justification:**

Thanks the authors for the response. I will keep my score as "accept".

**Limitations:**

1. It would be better to further address the limitations due to geodesic transport map.
2. It would be better to discuss the potential of efficient implementation of FGBoost, given that existing gradient boosting algorithms are know for its highly optimized implementations for fast training.

**Paper Formatting Concerns:**

No.

**Quality:**

4

**Strengths And Weaknesses:**

Strenghts:
1. As a powerful out-of-the-shelf machine learning algorithm, gradient boosting has been applied to many regression/classification/ranking problems. It is definitely quite valuable to extend gradient boosting to more diverse output spaces. In this case, a geodesic space.
2. Gradient boosting (with decision trees) has been always taken as black-box machine learning tools without prior knowledge applied in the algorithm. However, the results in Appendix H show the potential advantage of explicit geometry awareness through regression in the graph space, over simple vectorized numerial representations of graph with existing gradient boosting algorithm. I personally think this is interesting if the advantage can be more widely and thoroughly verified.
3. The approach lays on solid theoretical foundations. The adaption from traditional gradient boosting is natural and clear.
4. The paper is clearly written.

Weknesses:
1. Some details are missing, i.e., how the trees are constructed in FGBoost.
2. It would be better if FGBoost could be evaluated over more real-world applications, as well as comparing with traditional gradient boosting approaches.
3. The requirement of geodesic transport map may limit the application of the algorithm. For example, the geodesic transport map of distributions requires the calculation of quantiles, which is not trivial for most of the probability distributions.
4. Calculation of Frechet mean is more expensive than calculation of mean of scalar values in traditional gradient boosting. Thus FGBoost  could be more expensive. It would be good to think how to implement FGBoost efficiently. Especially when facing different metric spaces, the definition of distance can be very different. It can be hard to propose a unified library or framework which can handle all metric spaces efficiently.

---

> ### Author Rebuttal · Authors · 2025-07-28
>
> We are very grateful for your thorough review and positive assessment. We are particularly encouraged that you found the extension of gradient boosting to geodesic spaces to be "**quite valuable**," our theoretical foundations "**solid**," and the paper "**clearly written**." Your suggestion about highlighting the advantages of geometry awareness is a key insight that we appreciate.
>
> We thank you for the opportunity to clarify the missing details and discuss the limitations you raised. Below, we address each of your points.
>
> ---
>
> > Tree construction
>
> You raised an important point about missing details on how the trees are constructed. We will add a dedicated paragraph to the main text to make this procedure explicit.
>
> The tree is constructed greedily, similar to a standard CART algorithm. At each node, the algorithm searches for the optimal split (i.e., the feature and threshold) that maximizes the reduction in the overall loss function. Specifically:
> 1. For every potential split, the data is partitioned into two child nodes.
> 2. For each proposed child node, we compute a single representative geodesic, which is the Fréchet mean of the pseudo-residuals ($\gamma_{\hat{Y}_i^{k-1}, Y_i}$) of all samples falling into that node.
> 3. We then calculate the total Mean Squared Error (MSE) that would result from this split. The MSE is the sum of squared distances between the true observations $Y_i$ and the newly updated predictions. A sample's newly updated prediction is its current prediction updated with the representative geodesic from the child node it belongs to.
> 4. The algorithm selects the split that results in the lowest total MSE. This process is repeated recursively.
>
> This procedure is conceptually identical to standard regression tree construction, which chooses splits to minimize MSE. The key difference is that our "MSE" is defined by the metric of the geodesic output space, as per the loss function in lines 184-186.
>
> ---
>
> > Expanding real-world evaluation
>
> Thank you for your suggestion. We agree that including more real-world datasets would strengthen the empirical evaluation of the proposed method. In addition to the two real-world datasets presented in the main text (**human mortality distributional data** and **New York taxi traffic network data**), we present a third dataset in Appendix A.2, where we demonstrate the application of FGBoost for **emotional well-being compositional data**. To further enhance the empirical evaluation, we analyzed a fourth real-world dataset from the **National Health and Nutrition Examination Survey (NHANES)** 2005–2006, which we will include in the revision. NHANES combines interviews and physical exams to assess health and nutrition. During this cycle, participants aged six and older wore an ActiGraph 7164 accelerometer for seven days, recording physical activity intensity in 1-minute epochs, with values ranging from 0 to 32,767 CPM.
>
> We modeled the distribution of physical activity intensity using demographic and health-related predictors. For each participant, we exclude physical activity intensity values above 1000 CPM or equal to zero, as zeros may correspond to various low-activity states (e.g., sleep or swimming). The remaining physical activity intensity values over the seven days are concatenated to form the empirical distribution of each participant's physical activity intensity. Similar analyses of physical activity intensity have been conducted in previous studies (Chang & McKeague, 2022; Lin et al., 2023).
>
> Predictors include 13 variables: gender, age, race/ethnicity, veteran status, education (college or above), income (≥$35,000), marital status, weight, height, BMI, thigh circumference, waist circumference, and upper arm length. We selected 200 participants with the most valid data to ensure reliable coverage and evaluated performance using 10-fold cross-validation.
>
> Results show FGBoost outperforms competitors in prediction accuracy, achieving the lowest error (0.0544) and consistently low standard error; see lines 232-236 for the comparison methods.
>
> _Table: Average mean squared prediction errors and standard deviations (in parentheses)._
> |FGBoost|GFR|SDR|IFR|FRF|RFWLLFR|
> |-|-|-|-|-|-|
> |0.0544 (0.0010)|0.0590 (0.0011)|0.0649 (0.0058)|0.0711 (0.0121)|0.0575 (0.0008)|0.0729 (0.0028)|
>
> ---
>
> > Requirement for geodesic transport maps
>
> We acknowledge that the need for a transport map is a key assumption. However, for many practical cases, this is not a major barrier. Regarding your specific example of distributional data, when working with empirical distributions constructed from observations, the transport map relies on **empirical quantile functions**. These are **simple step functions** and are computationally straightforward. For the broader class of Riemannian manifolds, the transport map is well-defined as parallel transport. We will expand the discussion in the appendix to cover these practical considerations.
>
> ---
>
> > Computational efficiency and unified framework
>
> We agree that computational efficiency is important. Our implementation is designed to exploit closed-form solutions where possible and use efficient iterative methods otherwise, while maintaining a modular structure. In FGBoost, the metric-specific operations (distance evaluation and geodesic calculation) are separated from the core boosting procedure, which remains unchanged.
>
> While computing Fréchet means introduces additional cost compared to scalar averaging in standard boosting, this step is not a major bottleneck in many common settings:
> - Closed-form or low-cost computation:
>   - **Univariate distributions with the 2-Wasserstein metric**: The Fréchet mean is the $L^2$ average of quantile functions.
>   - **SPD matrices with power metrics**: Computed using the Fréchet mean in the power-transformed space, i.e., by mapping matrices through the power transform, averaging in that space, and applying the inverse transform.
>   - **Networks with the Frobenius metric**: Computed as the Fréchet mean under the Frobenius norm, which reduces to the arithmetic mean of graph Laplacians.
> - Iterative methods for complex spaces: When a closed form is unavailable, established algorithms such as the **proximal point method** (Bacák, 2014) are efficient and converge quickly under convexity.
>
> We acknowledge that FGBoost is inherently more computationally intensive than scalar-based boosting because of metric and Fréchet mean calculations. To ensure transparency, we report training time comparisons with other non-Euclidean methods in Appendix J.
>
> Developing a fully optimized, unified library for all metric spaces is an ambitious engineering task, similar to the evolution of frameworks like XGBoost and LightGBM from the original gradient boosting concept. Our contribution here is to provide a general algorithmic framework and demonstrate its feasibility across multiple spaces. Creating highly optimized implementations for specific domains is a promising direction for future work, and we will note this in the Limitations section.
>
> ---
>
> > SHAP values (Fig. 2)
>
> This is a sharp observation. In our adaptation of SHAP values for a geodesic space, we replace the standard (signed) difference in predictions with the **distance** between predictions, i.e., $d\big(f_{S \cup \{j\}}(\mathbf{x}_{S \cup \{j\}}), f_S(\mathbf{x}_S)\big)$ in lines 547-548. Since distance is always a non-negative quantity, the resulting feature contributions (our SHAP values) are also always non-negative. Figure 2 thus plots the raw SHAP values, which represent the magnitude of each feature's impact. We will add a sentence to Appendix C to explicitly clarify this.
>
> ---
>
> > Comparison with multi-output gradient boosting
>
> Thank you for this excellent suggestion and for pointing us to the relevant reference (Iosipoi & Vakhrushev, 2022). We implemented SketchBoost, a recent multi-output gradient boosting method, and compared it with FGBoost and the coordinate-wise XGBoost baseline under the same network simulation setup (see Appendix H). Since SketchBoost supports multi-output prediction, we vectorized the strict upper triangular entries of each graph Laplacian into a single vector-valued response and trained the model using default settings.
>
> The table below reports average mean squared prediction errors (with standard deviations) over 500 Monte Carlo replications. FGBoost consistently outperforms both XGBoost and SketchBoost across all sample sizes. While SketchBoost improves over XGBoost by leveraging joint modeling, its advantage appears only at larger sample sizes and it remains inferior to FGBoost. These results suggest that preserving the intrinsic non-Euclidean geometry of the output space, as FGBoost does, is critical for accurately capturing network structure.
>
> _Table: Average mean squared prediction errors and standard deviations (in parentheses) for network outputs._
> |$n$|FGBoost|XGBoost|SketchBoost|
> |-|-|-|-|
> |100|13.644 (3.140)|15.234 (3.319)|15.391 (2.825)|
> |200|10.531 (3.371)|11.989 (2.686)|12.162 (2.421)|
> |500|6.912 (1.950)|9.035 (2.145)|8.887 (1.869)|
> |1000|5.471 (1.481)|7.096 (1.703)|6.793 (1.443)|
>
> ---
>
> **References**
> - Bacák, M., 2014. Computing medians and means in Hadamard spaces. *SIAM Journal on Optimization*, 24(3), pp.1542-1566.
> - Iosipoi, L. and Vakhrushev, A., 2022. SketchBoost: Fast gradient boosted decision tree for multioutput problems. *Advances in Neural Information Processing Systems*, 35, pp.25422-25435.
> - Lin, Z., Kong, D. and Wang, L., 2023. Causal inference on distribution functions. *Journal of the Royal Statistical Society, Series B (Statistical Methodology)*, 85(2), pp.378-398.
> - Chang, H.W. and McKeague, I.W., 2022. Empirical likelihood-based inference for functional means with application to wearable device data. *Journal of the Royal Statistical Society, Series B (Statistical Methodology)*, 84(5), pp.1947-1968.

---

> > ### Comment · Reviewer_HbNk · 2025-08-04
> >
> > Thanks for the thorough response to the questions. The additional comparison with multi-output GBDT is quite supportive.

---

> > > ### Author Response · Authors · 2025-08-04
> > >
> > > Thank you for your thoughtful feedback and for acknowledging the additional comparison with multi-output GBDT. We appreciate your engagement and are glad the clarifications addressed your questions. We will ensure that these details and comparisons are clearly reflected in the revised manuscript to strengthen the contribution.

---

### Decision · Program_Chairs · 2025-09-17

**Decision:**

Accept (poster)

**Comment:**

The paper proposed a gradient boosting procedure with is adapted to non Euclidean output spaces (geodesic metric spaces). Experimental validation indicates that the method performs favorably compared to competitors. The reviewers were initially enthusiastic, and the rebuttal allowed providing some clarifications. Please integrate all the suggested changes (especially regarding the tree fitting procedure) to the revised version.